# Online Laplacian-Based Representation Learning in Reinforcement Learning

**Maheed H. Ahmed** [1]   **Jayanth Bhargav** [1]   **Mahsa Ghasemi** [1]

## Abstract

Representation learning plays a crucial role in reinforcement learning, especially in complex environments with high-dimensional and unstructured states. Effective representations can enhance the efficiency of learning algorithms by improving sample efficiency and generalization across tasks. This paper considers the Laplacian-based framework for representation learning, where the eigenvectors of the Laplacian matrix of the underlying transition graph are leveraged to encode meaningful features from raw sensory observations of the states. Despite the promising algorithmic advances in this framework, it remains an open question whether the Laplacian-based representations can be learned online and with theoretical guarantees along with policy learning. We address this by formulating an online optimization approach using the Asymmetric Graph Drawing Objective (AGDO) and analyzing its convergence via online projected gradient descent under mild assumptions. Our extensive simulation studies empirically validate the convergence guarantees to the true Laplacian representation. Furthermore, we provide insights into the compatibility of different reinforcement learning algorithms with online representation learning.

## 1. Introduction

Representation learning is an important part of machine learning that involves learning compact and useful representations of data. The quality of these representations significantly impacts the performance and efficiency of machine learning algorithms (Bengio et al., 2013). In reinforcement learning (RL), agents often deal with complex environments characterized by high-dimensional and unstructured states.

This makes representation learning important for discovering and encoding meaningful features from raw sensory inputs. The main goal of RL is to learn an optimal strategy (policy) that maps each state to an action, aiming to maximize the expected reward based on the dynamics of the environment. Learning a good representation can improve the sample efficiency of value-function approximation algorithms (Farebrother et al., 2023), a major family of RL algorithms, and enhance generalizations across different tasks (Yuan & Lu, 2022). In addition, representation learning has found applications in reward shaping (Wu et al., 2018), learning options with larger coverage (Machado et al., 2017a; Jinnai et al., 2019; Chen et al., 2024), zero-shot learning (Touati et al., 2022), and transfer learning (Gimelfarb et al., 2021; Barreto et al., 2017).

A graph representation is often used to learn a representation, i.e., a low-dimensional embedding, of the states (Mahadevan & Maggioni, 2007; Wu et al., 2018). States of an environment can be viewed as nodes of a graph, and the transition probability between states under a given policy can be viewed as weighted edges between these nodes. States that are closely connected in the graph are expected to have similar representations in the embedding space. One representation that retains this property is the eigenvectors of the graph Laplacian. Formally, the $d$-eigenvectors of the graph Laplacian corresponding to the $d$-smallest eigenvalues are used to construct an embedding function that maps a state to a vector in $\mathbb{R}^d$. We refer to those $d$-eigenvectors as the $d$-smallest eigenvectors for the remainder of this paper.

Constructing the graph and performing eigendecomposition on the Laplacian is only feasible in the tabular settings where the number of states is small. Therefore, Wu et al. (2018) proposed a scalable method to compute the smallest eigenvectors by solving an unconstrained version of the graph drawing objective (Koren, 2005) which is suitable for large and continuous state-spaces. However, the graph drawing objective does not have a unique minimizer, rather the rotations of the smallest eigenvectors are also its minimizers. To tackle this challenge, Wang et al. (2021) propose the generalized graph drawing objective which breaks the symmetry and only has the smallest eigenvectors as a unique minimizer. Gomez et al. (2023) show that under gradient descent dynamics, the unconstrained version of the generalized graph drawing objective has permutations of the small-

[1] Electrical and Computer Engineering, Purdue University, West Lafayette, IN 47907, USA. Correspondence to: Maheed H. Ahmed <ahmed237@purdue.edu>.

*Proceedings of the 42nd International Conference on Machine Learning*, Vancouver, Canada. PMLR 267, 2025. Copyright 2025 by the author(s).

est eigenvectors as equilibrium points. They propose the augmented Lagrangian Laplacian objective (ALLO) which has the smallest eigenvectors and the corresponding eigenvalues as the stable equilibrium under stochastic gradient descent-ascent dynamics.

The Laplacian-based representation can be computed or learned for a given policy according to its induced Markov chain. However, in RL the policy updates during the training phase as new data comes in, which will in turn necessitate recomputation of the representation. To avoid this complexity, in practice, the Laplacian-based representation is learned for a uniformly random policy in a pretraining phase and then used throughout training. Nevertheless, that fixed representation may not be effective for the policies encountered during RL. Recently, Klissarov & Machado (2023) showed that learning the representation in an online manner while simultaneously updating the policy can improve exploration and increase the total reward. In Figure 1, we illustrate an example, comparing the representations of a uniform policy and a non-uniform policy, that further underscores the need for adapting the representation. The non-uniform policy shows that some cells, despite being far from the target in terms of Euclidean distance, are actually closer in the embedding space than neighboring cells. This suggests that using the current representation to design rewards could offer a better signal for improving the policy. Klissarov & Machado (2023) proposed online deep Laplacian-based options for temporally extended exploration where a set of policies (also known as options) are trained to select exploratory actions using an estimated Laplacian representation of the current overall policy. They provide an extensive empirical analysis of how learning options while updating the representation increase the received rewards; however, the theoretical analysis of online representation learning while updating the policy has remained an open question.

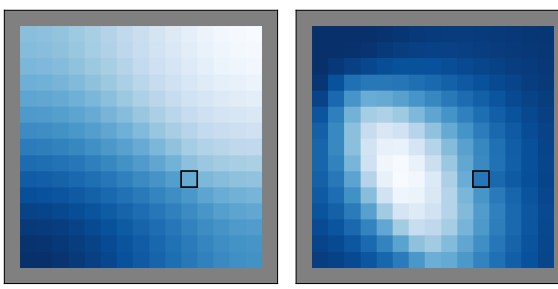

Figure 1: The Laplacian representation of a uniform policy (left) and a non-uniform policy (right). The color represents the entry corresponding to each state in the second eigenvector of the Laplacian. The bordered cell is the target.

Motivated by this, we derive a framework for online learning of the Laplacian representation. We summarize our

contributions below:

- We introduce the Asymmetric Graph Drawing Objective (AGDO), a simplified version of ALLO that eliminates the need for dual variables.

- We prove that under gradient descent dynamics, the only stable equilibrium for AGDO is the set of d-smallest eigenvectors.

- We theoretically and empirically establish that optimizing the online version of AGDO converges to a stationary point under the assumption of bounded drift.

## 2. Literature Review

In this section, we review existing studies and research directions in representation learning for reinforcement learning, focusing on topics closely linked to this study.

**Proto-Value Functions.** Mahadevan (2005) introduced proto-value functions, a set of basis functions that are independent of the reward function. These functions are defined as the eigenfunctions of the normalized Laplacian of the graph generated by a random walk over the state space. This representation has been demonstrated to reduce the number of samples required for training linear value function approximators (Mahadevan & Maggioni, 2007). The process of generating the graph involves collecting samples from the environment and connecting neighboring states with edges. However, this method does not adequately account for the stochastic nature of transitions and requires a discrete state space. In continuous state settings, Mahadevan (2012) proposes using the Nyström method to interpolate the values of eigenfunctions at unseen states based on visited states. Additionally, Xu et al. (2014) suggests enhancing representative state selection by applying K-means clustering to collected samples and constructing a graph from the resulting centroids.

**Laplacian Representation Using the Graph Drawing Objective.** Wu et al. (2018) formulated a linear operator that represents the graph over the state-space generated by a fixed policy, capturing the stochastic nature of transitions, and is applicable to continuous state spaces. They demonstrated that obtaining the eigenfunctions of the graph Laplacian, typically solved via the graph drawing objective (Koren, 2005), can be achieved through stochastic optimization using collected samples without explicitly constructing the graph. Additionally, they illustrated a method to recover these eigenfunctions up to orthonormal rotation by training a neural network. For precise eigenfunction recovery, Wang et al. (2021) introduced the generalized graph drawing objective, which breaks the symmetry inherent in

the traditional graph drawing objective. Despite the constrained generalized objective ensuring the uniqueness of the global minimizer being the Laplacian eigenfunctions, Gomez et al. (2023) demonstrated that stochastic optimization using the unconstrained objective—employed in neural network training—does not necessarily converge to these eigenfunctions. Consequently, they proposed the augmented Lagrangian Laplacian objective, which has the eigenvectors of the Laplacian as the unique stable equilibrium. Other equilibrium points correspond to permutations of the eigenvectors.

Learning the Laplacian representation with any of these objectives is conducted under a fixed policy, typically a uniformly random policy in practice. Klissarov & Machado (2023) introduced online deep covering eigenoptions (DCEO), an online algorithm that concurrently learns the Laplacian representation and options (Sutton, 1998), a well-established formulation of temporally extended actions in Markov Decision Processes (MDPs). They demonstrated that the online version of DCEO achieves performance comparable to a two-stage variant of the algorithm, where the representation is learned under a fixed uniform policy, with both approaches outperforming other baselines in multiple benchmarks.

**Successor Features.** The deep successor representation, introduced by Kulkarni et al. (2016) as an extension of the successor representation (Dayan, 1993), decomposes the value function into a successor feature function and a reward predictor function. The successor function encodes the discounted expected value of representations of all future states within a given horizon. Leveraging concepts from TD learning and Deep Q-networks (Mnih et al., 2015), both the representation and the successor feature function can be learned simultaneously with neural networks. Successor features have found diverse applications, such as sub-goal states generation in sparse reward environments (Kulkarni et al., 2016), transfer learning (Barreto et al., 2017; Gimelfarb et al., 2021), and options discovery (Machado et al., 2017b; 2023). Notably, Machado et al. (2017b) demonstrated a connection between the eigenvalues and eigenvectors of the successor representation matrix and the eigenvalues and eigenvectors of the normalized Laplacian defined as proto-value functions.

**Contrastive Learning in Reinforcement Learning.** Contrastive learning is a machine learning method used for learning representations that distinguish between similar and dissimilar pairs of data points using a contrastive loss function. Formally, an encoder is tasked with mapping data points to a latent representation where similar points are closely positioned in the latent space. For instance, Laskin et al. (2020) introduced the contrastive unsupervised rep-

resentations for reinforcement learning algorithm, where they train an encoder network using a contrastive loss with pairs of images randomly augmented from the same source image. The learned representation is subsequently utilized to train a deep reinforcement learning agent. Furthermore, augmented temporal contrast was developed by Stooke et al. (2021), which involves selecting similar sample pairs from samples that are separated by a short time distance. This approach is closely related to the Laplacian approach to representation learning, as states that are connected in the graph have a higher probability of appearing in consecutive samples than disconnected states.

In this work, we focus on extending the Laplacian-based representation learning, which has been shown in recent literature to be effective in learning options with high coverage Machado et al. (2017a); Jinnai et al. (2019); Klissarov & Machado (2023); Chen et al. (2024), to the online setting. While empirical results, such as those by Klissarov & Machado (2023), have demonstrated that online representation learning is effective and promising, a thorough theoretical analysis of the convergence and accuracy of these learned representations in the online setting is still lacking. Therefore, our work seeks to address this gap by developing a theoretical framework that ensures the stability and accuracy of Laplacian representations in an online learning context.

## 3. Preliminaries

In this section, we provide the necessary background to introduce the problem and present the proposed formulation and its theoretical analysis. We begin by introducing Markov decision processes within the context of reinforcement learning. Next, we highlight the closely related, existing methods of learning the Laplacian representation.

**Notation** We use $\langle v, u \rangle$ to denote the dot product between two vectors $v$ and $y$. For a vector $\mathbf{x}$, the $L_2$ norm, denoted $\|\mathbf{x}\|$, is defined as $\|\mathbf{x}\| = \sqrt{\sum_i |x_i|^2}$. The $L_2$ norm of a matrix, is defined as $\|\mathbf{A}\| = \sup_{\mathbf{x} \neq \mathbf{0}} \frac{\|A\mathbf{x}\|}{\|\mathbf{x}\|}$ and is equivalent to the spectral norm defined as the largest singular value of the matrix. Finally, the $L_\infty$ norm, denoted $\|\mathbf{A}\|_\infty$, is the maximum absolute row sum of the matrix, i.e., $\|\mathbf{A}\|_\infty = \max_i \sum_j |a_{ij}|$.

**Reinforcement Learning.** In the reinforcement learning setting, an agent interacts with an environment, which is modeled as a Markov decision process (MDP). A reward agnostic MDP is represented by the tuple $(\mathcal{S}, \mathcal{A}, \mathcal{T}, \mu_0)$ where $\mathcal{S}$ is the finite state space, $\mathcal{A}$ is the finite actions space, $\mathcal{T} : \mathcal{S} \times \mathcal{A} \to \Delta(\mathcal{S})$ is the transition probability, and $\mu_0 \in \Delta(\mathcal{S})$ is the initial state probability distribution. We consider the environment to be reward-agnostic and

that the agent has a policy $\pi : \mathcal{S} \to \Delta(\mathcal{A})$ from which actions are samples each time step. The policy induces a Markov chain from the MDP defined by the transition probability $P^\pi$ where $P^\pi(s, s') = \mathbf{P}(s_{t+1} = s' | s_t = s, \mathcal{T}, \pi) = \sum_{a \in \mathcal{A}} \pi(a|s) \mathcal{T}(s'|s, a)$. We assume that the induced Markov chain has a unique stationary distribution $\rho^\pi \in \Delta(\mathcal{S})$. We formally define this in Assumption 4.1.

**Laplacian Representation.** A graph is defined by a set of nodes $\mathcal{V}$ and an adjacency matrix $W \in \mathbb{R}^{|\mathcal{V}| \times |\mathcal{V}|}$. For two nodes $\nu, \nu'$, $W_{\nu, \nu'}$ is non-zero if and only if there exists an edge from $\nu$ to $\nu'$. The Laplacian matrix $L$ is defined as $L = D - W$ where the degree matrix $D$ is a diagonal matrix with $D_{\nu, \nu} = \sum_{j=1}^{|\mathcal{V}|} W_{\nu, j}$. The Laplacian encodes a lot of useful information about the underlying graph. For example, the second to the smallest eigenvalue also known as the Fiedler value determines the algebraic connectivity of the graph (Fiedler, 1973).

In the tabular setting, under a fixed policy $\pi$, an MDP can be represented as a graph, where $\mathcal{V} = \mathcal{S}$ and the adjacency matrix $W^\pi$ is defined as $f(P^\pi)$ where $f$ maps $P^\pi$ to a symmetric matrix. More generally, consider the following formulation given by Wu et al. (2018):

- A Hilbert Space $\mathcal{H}^\pi$ is $\mathbb{R}^{|\mathcal{S}|}$ with the inner product between two elements $u, v \in \mathcal{H}^\pi$ defined as $\langle u, v \rangle_{\mathcal{H}^\pi} = \sum_{s \in \mathcal{S}} u(s) v(s) \rho^\pi(s)$.

- A linear operator $A : \mathcal{H}^\pi \to \mathcal{H}^\pi$ is defined as $Au(s) = \sum_{s' \in \mathcal{S}} A(s, s') u(s') \rho^\pi(s')$.

- The self adjoint operator $W^\pi : \mathcal{H}^\pi \to \mathcal{H}^\pi$ is defined as

$$W^\pi(s, s') = \frac{1}{2} \frac{P^\pi(s, s')}{\rho^\pi(s')} + \frac{1}{2} \frac{P^\pi(s', s)}{\rho^\pi(s)} \quad (1)$$

- The Laplacian $L^\pi$ is defined as $L^\pi = \mathbf{I} - W^\pi$.

- With a slight abuse of notation we define $A_{\rho^\pi} : \left( \mathbb{R}^{|\mathcal{S}|}, \langle ., . \rangle \right) \to \left( \mathbb{R}^{|\mathcal{S}|}, \langle ., . \rangle \right)$ as a matrix whose entries are defined as $A_{\rho^\pi}(s, s') = A(s, s') \rho^\pi(s')$ for some operator $A : \mathcal{H}^\pi \to \mathcal{H}^\pi$. Note that for a vector $u \in \mathbb{R}^{|\mathcal{S}|}$ the matrix multiplication $A_{\rho^\pi} u$ is equivalent to $Au$.

We denote the $d$-smallest eigenvectors of $L^\pi$ as $e_1^\pi, e_2^\pi, \ldots, e_d^\pi$. The Laplacian embedding function $\phi^\pi : \mathcal{S} \to \mathbb{R}^d$ embeds a state $s$ to the $d$-dimensional vector whose $i$-th element correspond to the $s$-th element of $e_i^\pi$, i.e. $\phi(s) = [e_1^\pi[s], e_2^\pi[s], \ldots, e_d^\pi[s]]^\intercal$.

**Learning the Laplacian Representation.** Optimizing the graph drawing objective (GDO) (Koren, 2005) retrieves the

smallest $d$-eigenvectors up to orthonormal rotation. The graph drawing objective is defined as

$$\min_{u \in \mathbb{R}^{d|\mathcal{S}|}} \quad \sum_{i=1}^{d} \langle u_i, L^\pi u_i \rangle; \quad (2)$$
$$\text{s.t.} \quad \langle u_j, u_k \rangle = \delta_{jk}, \quad 1 \le k, j \le d,$$

where $\delta_{jk}$ is the Kronecker delta. The unconstrained approximation of GDO is defined as

$$\min_{u \in \mathbb{R}^{d|\mathcal{S}|}} \sum_{i=1}^{d} \langle u_i, L^\pi u_i \rangle + b \sum_{j=1}^{d} \sum_{k=1}^{d} \left( \langle u_j, u_k \rangle - \delta_{jk} \right)^2, \quad (3)$$

where $b$ is a hyper-parameter.

One advantage of using the graph drawing objective is that the unconstrained approximation of the graph drawing objective can be optimized by stochastic gradient descent using samples collected from the environment without constructing the graph or the Laplacian (Wu et al., 2018). Formally, if the inner product is defined in terms of $\rho^\pi$, the loss can be defined as $\sum_{i=1}^{d} \langle u_i, L^\pi u_i \rangle_{\mathcal{H}^\pi} = \mathbb{E}_{s \sim \rho^\pi, s' \sim P^\pi(.|s)} [\sum_{i=1}^{d} (u_i(s) - u_i(s'))^2]$.

The generalized graph drawing objective proposed by Wang et al. (2021) breaks the symmetry in the graph drawing objective and has the set of the smallest $d$-eigenvectors as a unique minimizer.

The generalized graph drawing objective (GGDO) is defined as

$$\min_{u \in \mathbb{R}^{d|\mathcal{S}|}} \quad \sum_{i=1}^{d} c_i \langle u_i, L^\pi u_i \rangle \quad (4)$$
$$\text{such that} \quad \langle u_j, u_k \rangle = \delta_{jk}, \quad 1 \le k, j \le d$$

and the unconstrained approximation of GGDO is defined as

$$\min_{u \in \mathbb{R}^{d|\mathcal{S}|}} \sum_{i=1}^{d} c_i \langle u_i, L^\pi u_i \rangle + $$
$$b \sum_{j=1}^{d} \sum_{k=1}^{d} \min(c_j, c_k) \left( \langle u_j, u_k \rangle - \delta_{jk} \right)^2 \quad (5)$$

The unconstrained GGDO is guaranteed to have a unique equilibrium point only in the limit $b \to \infty$. However, for other values, rotations of the smallest $d$-eigenvectors can still be an equilibrium point. The augmented Lagrangian Laplacian objective (ALLO) suggested by (Gomez et al., 2023) is a dual objective that has a unique stable equilibrium point of the smallest $d$-eigenvalues and the corresponding smallest $d$-eigenvectors. Other unstable equilibrium points

correspond to permutations of the eigenvectors and eigenvalues. The ALLO is defined as follows

$$
\max_{\beta} \min_{u \in \mathbb{R}^{d|\mathcal{S}|}} \sum_{i=1}^{d} \langle u_i, L^{\pi} u_i \rangle +
$$

$$
\sum_{j=1}^{d} \sum_{k=1}^{j} \beta_{jk} \left( \langle u_j, [\![u_k]\!] \rangle - \delta_{jk} \right) + \quad (6)
$$

$$
b \sum_{j=1}^{d} \sum_{k=1}^{j} \left( \langle u_j, [\![u_k]\!] \rangle - \delta_{jk} \right)^2
$$

where $[\![.]\!]$ is the stop gradient operator, and whatever is inside the operator is treated as a constant when computing the gradient. The stop gradient operator has the same effect on breaking the symmetry as the introduction of the constant hyper-parameters in GGDO.

# 4. Online Learning of the Laplacian Representation

We first formulate the problem of learning the Laplacian representation while simultaneously updating the policy. We then provide theoretical bounds for the convergence of the learned representation.

## 4.1. Problem Definition

We formulate the problem of learning the Laplacian representation while the policy is updating as a sequence of Asymmetric GDOs (AGDOs) varying in time. To break the symmetry in GDO we apply the stop gradient operator similar to ALLO. We assume the policy $\pi_0$ is initialized randomly and some learning algorithm updates the policy in $T$ discrete time steps producing a policy $\pi_t$ after each update. Learning the Laplacian representation can then be represented by the sequence of objectives as follows

$$
\min_{u \in \mathcal{C}^{(t)}} \mathcal{L}^{(t)}(u) = \min_{u \in \mathcal{C}^{(t)}} \sum_{i=1}^{d} \langle u_i, L^{(t)} u_i \rangle_{\mathcal{H}^{(t)}} +
$$

$$
b \sum_{j=1}^{d} \sum_{k=1}^{j-1} \left( \langle u_j, [\![u_k]\!] \rangle_{\mathcal{H}^{(t)}} \right)^2 + \quad (7)
$$

$$
\frac{b}{2} \sum_{i=1}^{d} \left( \langle u_i, u_i \rangle_{\mathcal{H}^{(t)}} - 1 \right)^2
$$

where $\mathcal{C}^{(t)} \subset \mathbb{R}^{d|\mathcal{S}|}$ is a convex and closed set. We write $L^{\pi_t}$ and $\mathcal{H}^{\pi_t}$ as $L^{(t)}$ and $\mathcal{H}^{(t)}$ for simpler notation. In addition, we assume that $b > 0$. We refer to this objective as the asymmetric graph drawing objective (AGDO).[1]

---

[1] Here we have a slightly different application of the stop gradient operator than the objective proposed by Gomez et al. (2023).

Note that for a fixed policy, AGDO is a special case of ALLO with $\beta = 0$. Another similarity between AGDO and ALLO is that AGDO can be viewed as solving ALLO with added regularization for the dual parameters $\beta$ with a regularization parameter $\Gamma$. Adding a regularization term $-\Gamma \sum_{j=1}^{d} \sum_{k=1}^{j} \frac{\beta_{jk}^2}{2}$ to (6) yields a closed form solution for maximization over $\beta$ with $\beta_{jk}^*(u) = \frac{\langle u_j, [\![u_k]\!] \rangle_{\mathcal{H}^{(t)}} - \delta_{jk}}{\Gamma}$. Substituting reduces (6) to

$$
\min_{u \in \mathbb{R}^{d|\mathcal{S}|}} \sum_{i=1}^{d} \langle u_i, L^{(t)} u_i \rangle_{\mathcal{H}^{(t)}} +
$$

$$
(b + \frac{1}{2\Gamma}) \sum_{j=1}^{d} \sum_{k=1}^{j} \left( \langle u_j, [\![u_k]\!] \rangle_{\mathcal{H}^{(t)}} - \delta_{jk} \right)^2 \quad (8)
$$

which is the same as ALLO ($\beta = 0$) with $b$ replaced with $b + \frac{1}{2\Gamma}$ which is also a constant hyperparameter.

We lay the assumptions for our theoretical analysis.

**Assumption 4.1.** For each policy $\pi_t$, the induced Markov chain is ergodic and has a unique stationary distribution with non-zero entries, i.e., $\min_t \min_{s \in \mathcal{S}} \rho^{\pi_t}(s) = \rho_{\min} > 0$.

**Assumption 4.2.** For two consecutive time steps $t$ and $t+1$, the policies $\pi_t$ and $\pi_{t+1}$ satisfy $\max_{s \in \mathcal{S}} \sum_{a \in \mathcal{A}} |\pi_t(a|s) - \pi_{t+1}(a|s)| \leq \delta_\pi^{(t)}$. Additionally, the bound $\delta_\pi^{(t)}$ on the policy drift satisfies $\sum_{t=0}^{T} \delta_\pi^{(t)} = \mathcal{O}(f(T))$ for some sublinear function $f$.

Assumption 4.1 guarantees that the induced Markov chain has a unique stationary distribution and the induced probability measure $\rho^{(t)}$ assigns a non-zero value to every state. Assuming a unique stationary distribution is a common assumption in the theoretical reinforcement learning literature (Melo et al., 2008; Even-Dar et al., 2009). Note that going from $\rho^{(t)}(s) = 0$ to $\rho^{(t+1)}(s) > 0$ is equivalent to adding a node to the graph which would make the dimensions of the spaces inconsistent. A more general assumption can be made that $\rho^{(t+1)}$ is absolutely continuous with respect to $\rho^{(t)}$, i.e. $\rho^{(t)}(s) = 0 \implies \rho^{(t+1)}(s) = 0$, in which case, the same analysis can be done to the set $\mathcal{S}' = \{s \in \mathcal{S} : \rho^{(t+1)}(s) \neq 0\}$.

Assumption 4.2 assumes the drift in the policy caused by the policy learning algorithm is bounded. This bounded drift assumption is valid for many policy learning algorithms in RL, such as trust region policy optimization (TRPO) (Schulman et al., 2015) and proximal policy optimization (PPO) (Schulman et al., 2017). In addition, we require the learning algorithm to converge to some policy such that the total drift

---

The penalty term for the norm of $u_i$ does not have the stop gradient operator which does not change the gradient but ensures the term is propagated to the Hessian for the stability analysis. We provide more discussion in A.2

is sub-linear in $T$. For example, two commonly used techniques in on-policy deep reinforcement learning—learning rate annealing and gradient clipping (Andrychowicz et al., 2020; Engstrom et al., 2020)—can ensure that the sequence of policy updates remains bounded while gradually decreasing over time.

## 4.2. Convergence Analysis of AGDO

We first define the function $g_{u_i}^{(t)} : \mathbb{R}^{d|\mathcal{S}|} \to \mathbb{R}^{|\mathcal{S}|}$, which is the gradient of (7) with respect to $u_i$ taking into account the stop gradient operator, as

$$g_{u_i}^{(t)}(u) = \left( 2L^{(t)}u_i + 2b\sum_{j=1}^{i-1} \langle u_i, [\![u_j]\!] \rangle_{\mathcal{H}^{(t)}} [\![u_j]\!] \right) \odot \rho +$$
$$(2b\left( \langle u_i, u_i \rangle_{\mathcal{H}^{(t)}} - 1 \right) u_i) \odot \rho$$
(9)

where $\odot$ is the Hadamard product. The vectors $u_i$ are updated using the update equation

$$u_i^{(t+1)} \leftarrow \text{Proj}_{\mathcal{C}^{(t)}}(u_i^{(t)} - \eta g_{u_i}^{(t)}(u^{(t)})) = u_i^{(t)} - \eta G_{u_i}^{(t)}(u^{(t)})$$
(10)

where $\eta > 0$ is the learning rate, $\text{Proj}_{\mathcal{C}^{(t)}}$ is the projection to $\mathcal{C}^{(t)}$, and $G_{u_i}^{(t)}$ is the gradient map defined as $G_{u_i}^{(t)}(u^{(t)}) = \frac{1}{\eta}(u_i^{(t)} - \text{Proj}_{\mathcal{C}^{(t)}}(u_i^{(t)} - \eta g_{u_i}^{(t)}(u^{(t)})))$.

We show in Lemma 4.3 that for a fixed policy, if $\mathcal{C}^{(t)} = \mathbb{R}^{d|\mathcal{S}|}$, the equilibrium points of performing gradient descent to minimize the function $\mathcal{L}^{(t)}$ defined in (7) correspond to permutations of the eigenvectors. We defer all detailed proofs to Appendix A.

**Lemma 4.3.** *If $\mathcal{C}^{(t)} = \mathbb{R}^{d|\mathcal{S}|}$, $u^{*(t)}$ is an equilibrium point of the objective $\mathcal{L}^{(t)}$ in (7) under gradient descent dynamics, iff $u_i^{*(t)} = e_{\sigma(i)}^{(t)} m_i$, and $\langle u_i^{*(t)}, u_i^{*(t)} \rangle_{\mathcal{H}^{(t)}} = m_i \left( 1 - \frac{\lambda_{\sigma(i)}^{(t)}}{b} \right)$ for some permutation $\sigma : \mathcal{S} \to \mathcal{S}$ where $m_i \in \{0, 1\}$, i.e. zero or more vectors $u_i^{*(t)}$ can be zero.*

This result is similar to Lemma 2 derived by Gomez et al. (2023) with the norm of the vectors being different and the fact that vectors can be zero. However, we show in Theorem 4.4 that only the identity permutation with non-zero vectors corresponds to a stable equilibrium under proper selection of hyperparameters.

**Theorem 4.4.** *The only stable equilibrium point from Lemma 4.3 minimizing the objective $\mathcal{L}^{(t)}$ in (7) under gradient descent dynamics is the one corresponding to the identity permutation with none of the vectors being zero, under an appropriate selection of the barrier coefficient $b$, if the highest eigenvalue multiplicity is 1.*

## 4.3. Convergence Analysis of Online AGDO

In this section, we present a theoretical analysis of the convergence of the online PGD algorithm. We first begin by establishing certain properties of the PGD algorithm.

We consider the case where the vectors $u_i^{(t)}$ are constrained such that their norm is bounded. We define $\mathcal{C}^{(t)}$ as $\mathcal{C}^{(t)} = \{u \in \mathbb{R}^{d|\mathcal{S}|} : \langle u_i, u_i \rangle \leq \frac{2}{\rho_{min}}\}$. This set has two interesting properties. First, it includes all equilibrium points for all $b > 1$ (as established in Lemma 4.3). Second, the gradient function $g^{(t)}$ defined in (9) is Lipchitz continuous over $\mathcal{C}^{(t)}$. The following result establishes this property.

**Proposition 4.5.** *The loss function $\mathcal{L}^{(t)}$ defined in (7) is $\alpha$-smooth with Lipschitz continuous gradient $g^{(t)}$ such that*

$$\|g^{(t)}(u) - g^{(t)}(u')\| \leq \alpha \|u - u'\| \qquad (11)$$

*for any $u, u' \in \mathcal{C}^{(t)}$ with $\alpha = 2 + \left( 2 + \frac{12+4d}{\rho_{min}} \right) b$.*

Next, we present some preliminary results, which we will later use in the convergence analysis. In Lemma 4.6, we characterize the drift in the policy-induced Markov chain, the Laplacian operator, and the loss function.

**Lemma 4.6.** *Under Assumptions 4.1 and 4.2, we have the following:*

*(a) $\|P^{(t+1)} - P^{(t)}\|_\infty \leq \delta_\pi^{(t)}$*

*(b) $\|\rho^{(t+1)} - \rho^{(t)}\|_\infty \leq \delta_\rho^{(t)} = \kappa^{(t)} \delta_\pi^{(t)}$*

*(c) $\|(\rho^{(t+1)} \otimes \mathbf{1}) \odot L_{\rho^{(t+1)}}^{(t+1)} - (\rho^{(t)} \otimes \mathbf{1}) \odot L_{\rho^{(t)}}^{(t)}\| \leq \delta_L^{(t)} = \sqrt{|\mathcal{S}|} \left( \delta_\pi^{(t)} + \delta_\rho^{(t)} \right) + \delta_\rho^{(t)}$*

*(d) $\left| \mathcal{L}^{(t+1)}(u) - \mathcal{L}^{(t)}(u) \right| \leq \delta_\mathcal{L}^{(t)} = \frac{2d\delta_L^{(t)}}{\rho_{min}} + \frac{8bd^2\delta_\rho^{(t)}}{\rho_{min}^2};$*
*$\forall u \in \mathcal{C}^{(t)}$*

*where $\kappa^{(t)}$ is a condition number on the induced Markov chain by $\pi^{(t)}$.*

Note that Lemma 4.6(b) follows directly from previous work on the perturbation analysis of stationary distributions of Markov chains (Haviv & Van der Heyden, 1984; Funderlic & Meyer Jr, 1986; Cho & Meyer, 2000). For example, Cho & Meyer (2000) gives the following condition number $\kappa^{(t)} = \frac{1}{2} \max_j \max_{i \neq j} \frac{m_{ij}}{m_{jj}}$, where $m_{ij}$ is the mean first passage time from state $i$ to state $j$ and $m_{jj}$ is the mean return time to state $j$. For other possible options of condition numbers, review the comparative study by Cho & Meyer (2001).

Finally, we show in Theorem 4.7 that running online projected gradient descent on AGDO achieves ergodic convergence.

---

**Algorithm 1** Online PGD of AGDO
---

1: **Input:** Initial policy $\pi_0$, learning rate $\eta$, initial vector $u^{(0)}$, policy learning algorithm $\mathcal{A}$
2: **for** $t = 1$ to $T$ **do**
3:     Interact with the environment and add transitions to the replay buffer
4:     $u_i^{(t+1)} \leftarrow \text{Proj}_{\mathcal{C}^{(t)}}(u_i^{(t)} - \eta g_{u_i}^{(t)}(u^{(t)}))$
5:     Get $\pi_t$ by updating $\pi_{t-1}$ using $\mathcal{A}$
6: **end for**

---

**Theorem 4.7.** *Under Assumptions 4.1 and 4.2, running Algorithm 1 on the sequence of losses as defined in (7) for $T$ time steps, with a constant learning rate $\eta = \dfrac{1}{\alpha}$, we have,*

$$\mathbb{E}_{t\sim Uniform\{1,2,\ldots,T\}}\left[\|G^{(t)}(u^{(t)})\|^2\right] \leq$$
$$\frac{2\alpha}{T}\left(\mathcal{L}^{(1)}(u^{(1)}) - \mathcal{L}^* + \sum_{t=1}^{T}\delta_{\mathcal{L}}^{(t)}\right) = \mathcal{O}\left(\frac{f(T)}{T}\right) \quad (12)$$

*where $\mathcal{L}^*$ is the minimum value $\mathcal{L}^{(t)}$ can take and $G^{(t)}$ refers to the concatenation of the gradient vectors $G_{u_i}^{(t)}$. Moreover, the OPGD algorithm (Algorithm 1) under the time-varying loss function ((7)) asymptotically converges to the critical point.*

## 5. Empirical Analysis

We evaluate the accuracy of the proposed method in the fixed policy setting and the online setting. We evaluate the importance of different components of the algorithm as well.

**Experiments Setup** We consider the grid world environments shown in Figure 4. For each experiment, a fixed target location is sampled uniformly at random, and the agent receives a reward of $+1$ if the agent reaches the location. For each instance of the environment, a fixed target is sampled uniformly at random at the beginning of the training process. At the start of each episode or when the agent reaches the target, the new agent location is sampled uniformly at random. We consider a maximum episode length of 1000 steps. The matrix $\hat{P}^{(t)}$, used to compute the Laplacian $\hat{L}^{(t)}$, is defined using a weighted sum between the actual $P^{(t)}$ and the initial distribution, as suggested by Wu et al. (2018) to handle episodic Markov Decision Processes (MDPs). To compute the true Laplacian representation, we perform eigen decomposition on the matrix $\hat{L}_{\rho^{(t)}}^{(t)}$, which is equivalent to applying the Laplacian operator in the space $\mathcal{H}^{(t)}$.

We follow the same setting as Gomez et al. (2023), where we set $d = 11$ and use the $(x, y)$ coordinates as input to the encoder network, a fully connected neural network with 3

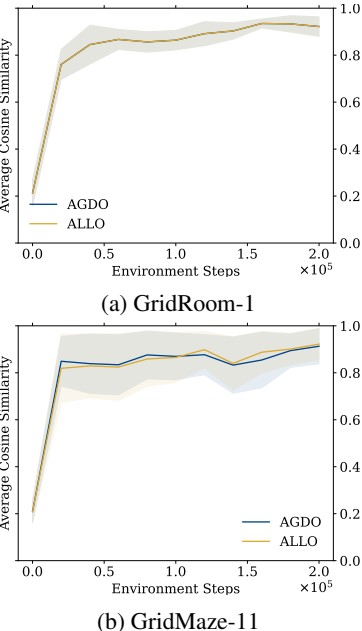

(a) GridRoom-1

(b) GridMaze-11

Figure 2: Average cosine similarity between the true Laplacian representation and the learned representation using AGDO and ALLO for a fixed uniform policy.

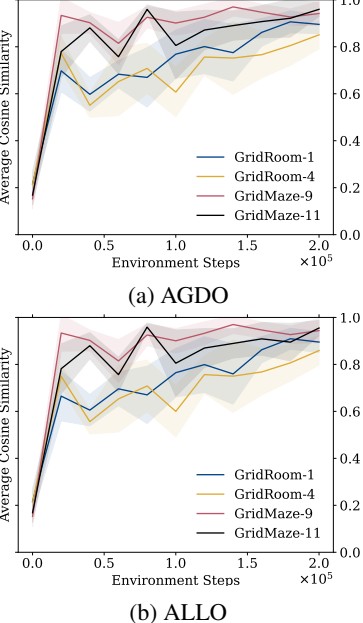

(a) AGDO

(b) ALLO

Figure 3: Average cosine similarity between the true Laplacian representation and the learned representation using AGDO and ALLO for a ppo policy.

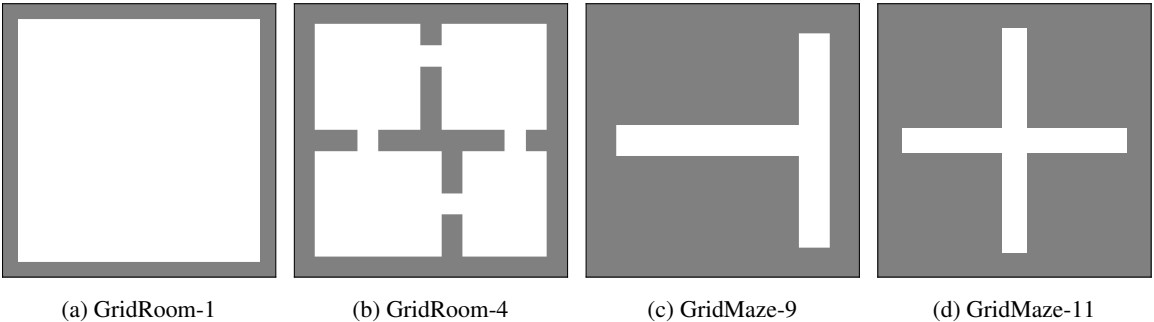

(a) GridRoom-1         (b) GridRoom-4         (c) GridMaze-9         (d) GridMaze-11

Figure 4: Environments tested in experiments where the grey areas are walls.

layers of size 256 each. We start training the encoder and the agent after collecting $10^4$ samples and run the experiment until $2 \times 10^5$ samples have been collected. We use a fixed value of 5 for the barrier coefficient. The encoder network is trained using an Adam optimizer with a learning rate of $10^{-3}$. For each collected sample, 10 batches are sampled to update the encoder. For training the agent, we use proximal policy optimization (PPO) (Schulman et al., 2017) as the training algorithm with an initial clipping parameter 0.2 unless otherwise specified. We add an entropy regularization term to discourage deterministic policies. To simulate assumption 4.2, we schedule the clipping parameter to decrease from 0.2 to 0.01 starting from step $10^5$ until the end of the training. For the full experimental setup, please refer to Appendix B. We provide an open-source implementation at `https://github.com/MaheedHatem/onl ine_laplacian_representation`. In all figures, we report the average cosine similarity of all dimensions of the eigenvectors averaged across 5 seeds with the 95% confidence interval highlighted.

**Eigenvalue Accuracy (Fixed Setting)** We start by comparing the performance of AGDO to ALLO in the fixed uniform policy setting. In Figure 2, we show that the average cosine similarity of AGDO and ALLO is almost identical for the same initial seeds. This result is similar to the analysis by Gomez et al. (2023) that showed that ALLO with $\beta = 0$ achieved similar results to ALLO.

**Eigenvalue Accuracy (Online Setting)** Figure 3 shows the results of optimizing both AGDO and ALLO in an online setting where the agent's policy is updated with the PPO loss. Similar to the fixed setting, the results of AGDO and ALLO are almost identical for the same set of seeds. In addition, for all environments, the average similarity trends upward as the training steps increase. For environments with a large number of states (GridRoom-1 and GridRoom-4) we notice that the accuracy is slightly lower at earlier stages of the training, which is coherent with our theoretical analysis (see Lemma 4.6 and Theorem 4.7) that the drift increases

with the number of states, resulting in slower convergence. However, this can be mitigated by imposing stricter bounds on the drift in the policy learning algorithm.

**Ablation Study** In this study, we aim to analyze three points: (1) the importance of the drift bound assumption, (2) the effect of the number of encoder update steps per sample collected, and (3) the effect of noise caused by sampling from the replay buffer when the policy was different.

To assert the importance of the bounded drift assumption, we compare running PPO with different initial clipping parameters, vanilla policy gradient (VPG) (Sutton et al., 1999), and deep Q-network (DQN) (Mnih et al., 2015). First note that VPG is equivalent to PPO without clipping. We can see in Figure 5a that the lower the clipping value is, i.e. the drift bound between policies is smaller, the higher the accuracy for the learned representation is. However, a small drift might affect the performance of the learned policy. In addition, for DQN the change in the policy distribution can be drastic for an $\epsilon-$greedy policy with a small $\epsilon$ whenever the Q-network changes the estimated optimal action in a state. As for the new estimated optimal action, the probability will shift from $\frac{\epsilon}{|\mathcal{A}|}$ to $1 - \epsilon$. This explains why the accuracy of the learned representation for DQN is much lower than the on-policy methods. We conclude that the bounded drift assumption is necessary for learning an accurate representation.

In Figure 5b, we analyze the effect of increasing the number of steps. We vary the number of update steps per sample between 1 and 20. While an increase in the number of steps is expected to enhance accuracy, our findings indicate that this is not observed. We hypothesize that this discrepancy is due to the presence of noise, caused by sampling from the replay buffer.

To confirm the previous hypothesis, we test in Figure 5c the effect of varying the replay buffer size. Recall that estimating the AGDO loss in (7) is done through sampling steps from the replay buffer. In the online setting, the buffer would include steps from previous policies with different

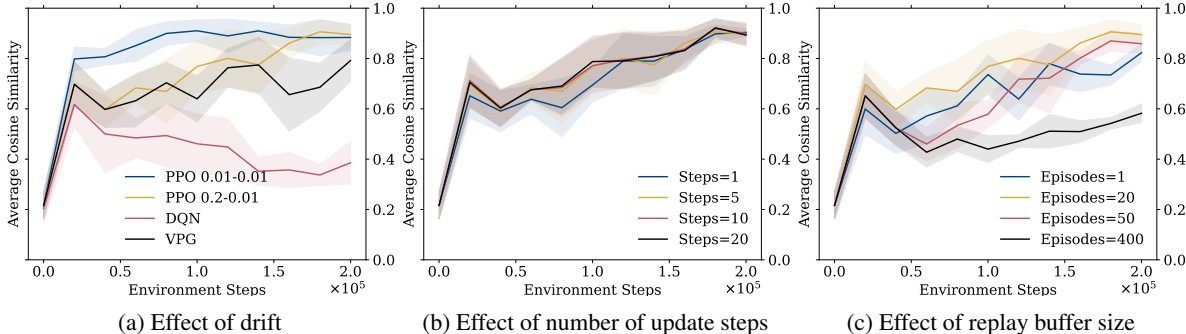

(a) Effect of drift      (b) Effect of number of update steps      (c) Effect of replay buffer size

Figure 5: Analysis of different aspects of online AGDO. (a) The effect of bounded drift on the accuracy of the learned representation. (b) The effect of the number of update steps per sample collected. (c) The effect of the number of episodes in the replay buffer.

stationary and transition distributions which would introduce bias to our loss estimate. However, a small buffer size would also increase the variance of the estimate. This is confirmed by the results, as for a buffer that holds only one episode we see a worse performance than a buffer that holds 20 episodes. On the other hand, increasing the buffer size drastically also causes accuracy to drop as the samples used have a different distribution which can be seen for buffers with sizes 50 and 400.

## 6. Conclusion and Future Work

In this paper, we studied online Laplacian-based representation learning and demonstrated that it can be effectively integrated with reinforcement learning, enabling simultaneous updates of both representation and policy. Our theoretical analysis, under mild assumptions, shows that running the online projected gradient descent on the Asymmetric Graph Drawing Objective achieves ergodic convergence, ensuring that the learned representations are aligned with the underlying dynamics. Additionally, our empirical studies reinforce these findings and give insight into the compatibility of reinforcement learning algorithms with online representation learning.

Our work opens new avenues for enhancing representation learning in complex environments and lays out the assumptions needed for its success. Future research could investigate how online Laplacian representation learning integrates with various learning paradigms, such as linear value function approximations or options learning. Additionally, an important direction would be to explore its applicability in non-stationary settings, where the perceived transition dynamics change over time either due to changes in the environment or the interactions with other learning agents in the multi-agent scenario.

## Acknowledgements

This work was supported in part by the National Aeronautics and Space Administration under Grant 80NSSC24M0070.

## Impact Statement

This work focuses on efficient online representation learning in reinforcement learning. There are many potential societal consequences of our work, none of which we feel must be specifically highlighted here.

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

# A. Proofs

## A.1. Proof of Lemma 4.3

*Proof.* For an equilibrium point

$$g_{u_i}^{(t)}(u^{*(t)}) = \left( 2L^{(t)}u_i + 2b\sum_{j=1}^{i-1} \langle u_i, u_j \rangle_{\mathcal{H}^{(t)}} u_j + 2b\left( \langle u_i, u_i \rangle_{\mathcal{H}^{(t)}} - 1 \right) u_i \right) \odot \rho = 0,$$

and since $\rho_{\min} > 0$, we can divide each element of the vectors on both sides by $\rho(s)$ and we get,

$$g_{u_i}^{(t)}(u^{*(t)}) = 2L^{(t)}u_i + 2b\sum_{j=1}^{i-1} \langle u_i, u_j \rangle_{\mathcal{H}^{(t)}} u_j + 2b\left( \langle u_i, u_i \rangle_{\mathcal{H}^{(t)}} - 1 \right) u_i = 0,$$

We proceed by induction. For the base case with $i = 1$, we have

$$g_{u_i}^{(t)}(u^{*(t)}) = 2L^{(t)}u_1^{*(t)} + 2b\left( \langle u_1^{*(t)}, u_1^{*(t)} \rangle_{\mathcal{H}^{(t)}} - 1 \right) u_1^{*(t)} = 0$$

Hence, either $u_1^{*(t)} = e_{\sigma(1)}^{(t)}$; for some permutation $\sigma : \mathcal{S} \to \mathcal{S}$ and $-2\lambda_{\sigma(1)}^{(t)} = 2b\left( \langle u_1^{*(t)}, u_1^{*(t)} \rangle_{\mathcal{H}^{(t)}} - 1 \right)$ (i.e., $\langle e_{\sigma(1)}^{(t)}, e_{\sigma(1)}^{(t)} \rangle_{\mathcal{H}^{(t)}} = 1 - \frac{\lambda_{\sigma(1)}^{(t)}}{b}$ ) or $u_1^{*(t)} = 0$.

Suppose now that either $u_j^{*(t)} = e_{\sigma(j)}^{(t)}$ and $\langle e_{\sigma(j)}^{(t)}, e_{\sigma(j)}^{(t)} \rangle_{\mathcal{H}^{(t)}} = 1 - \frac{\lambda_{\sigma(j)}^{(t)}}{b}$ or $u_j^{*(t)} = 0$ for all $j < i$ then the gradient becomes

$$g_{u_i}^{(t)}(u^{*(t)}) = 2L^{(t)}u_i^{*(t)} + 2b\left( \langle u_i^{*(t)}, u_i^{*(t)} \rangle_{\mathcal{H}^{(t)}} - 1 \right) u_i^{*(t)} +$$

$$2b\sum_{j=1}^{i-1} \langle u_i^{*(t)}, e_{\sigma(j)}^{(t)} \rangle_{\mathcal{H}^{(t)}} e_{\sigma(j)}^{(t)} \mathbb{1}_{u_j^{*(t)} \neq 0} = 0$$

Since the eigenvectors form a basis, let $u_i^{*(t)} = \sum_{k=1}^{|S|} c_{ik} e_{\sigma(k)}^{(t)}$. The gradient then becomes

$$g_{u_i}^{(t)}(u^{*(t)}) = \sum_{k=1}^{|S|} \left( 2\lambda_{\sigma(k)}^{(t)} + 2b\left( \langle u_i^{*(t)}, u_i^{*(t)} \rangle_{\mathcal{H}^{(t)}} - 1 \right) \right) c_{ik} e_{\sigma(k)}^{(t)}$$

$$+ 2b\sum_{j=1}^{i-1} \langle u_i^{*(t)}, e_{\sigma(j)}^{(t)} \rangle_{\mathcal{H}^{(t)}} e_{\sigma(j)}^{(t)} \mathbb{1}_{u_j^{*(t)} \neq 0} = 0.$$

(13)

Since the eigenvectors form a basis, all coefficients must be zero. For $j < i$ and $u_j^{*(t)} \neq 0$, we have:

$$\left( 2\lambda_{\sigma(j)}^{(t)} + 2b\left( \langle u_i^{*(t)}, u_i^{*(t)} \rangle_{\mathcal{H}^{(t)}} - 1 \right) \right) c_{ij} + 2b\langle u_i^{*(t)}, e_{\sigma(j)}^{(t)} \rangle_{\mathcal{H}^{(t)}} = 0 \quad (14)$$

Now note that

$$c_{ij} = \frac{\langle u_i^{*(t)}, e_{\sigma(j)}^{(t)} \rangle_{\mathcal{H}^{(t)}}}{\langle e_{\sigma(j)}^{(t)}, e_{\sigma(j)}^{(t)} \rangle_{\mathcal{H}^{(t)}}}.$$

Equation 14 then becomes

$$\left( \frac{2\lambda_{\sigma(j)}^{(t)} + 2b\left( \langle u_i^{*(t)}, u_i^{*(t)} \rangle_{\mathcal{H}^{(t)}} - 1 \right)}{\langle e_{\sigma(j)}^{(t)}, e_{\sigma(j)}^{(t)} \rangle_{\mathcal{H}^{(t)}}} + 2b \right) \langle u_i^{*(t)}, e_{\sigma(j)}^{(t)} \rangle_{\mathcal{H}^{(t)}} = 0$$

Reordering the terms, we have:

$$\left( \frac{2\lambda_{\sigma(j)}^{(t)} + 2b \left( \langle u_i^{*(t)}, u_i^{*(t)} \rangle_{\mathcal{H}^{(t)}} - 1 + \langle e_{\sigma(j)}^{(t)}, e_{\sigma(j)}^{(t)} \rangle_{\mathcal{H}^{(t)}} \right)}{\langle e_{\sigma(j)}^{(t)}, e_{\sigma(j)}^{(t)} \rangle_{\mathcal{H}^{(t)}}} \right) \langle u_i^{*(t)}, e_{\sigma(j)}^{(t)} \rangle_{\mathcal{H}^{(t)}} = 0.$$

Substituting $\langle e_{\sigma(j)}^{(t)}, e_{\sigma(j)}^{(t)} \rangle_{\mathcal{H}^{(t)}} = 1 - \frac{\lambda_{\sigma(j)}^{(t)}}{b}$, we have:

$$\left( \frac{2b \langle u_i^{*(t)}, u_i^{*(t)} \rangle_{\mathcal{H}^{(t)}}}{\langle e_{\sigma(j)}^{(t)}, e_{\sigma(j)}^{(t)} \rangle_{\mathcal{H}^{(t)}}} \right) \langle u_i^{*(t)}, e_{\sigma(j)}^{(t)} \rangle_{\mathcal{H}^{(t)}} = 0,$$

which implies that either $\langle u_i^{*(t)}, e_{\sigma(j)}^{(t)} \rangle_{\mathcal{H}^{(t)}} = c_{ij} = 0$ or $\frac{2b \langle u_i^{*(t)}, u_i^{*(t)} \rangle_{\mathcal{H}^{(t)}}}{\langle e_{\sigma(j)}^{(t)}, e_{\sigma(j)}^{(t)} \rangle_{\mathcal{H}^{(t)}}} = 0$, but the second condition is only true if $u_i^{*(t)} = 0$ which implies that $\langle u_i^{*(t)}, e_{\sigma(j)}^{(t)} \rangle_{\mathcal{H}^{(t)}}$ is always zero. For $k \geq i$ in (13) or $u_k^{*(t)} = 0$

$$\left( 2\lambda_{\sigma(k)}^{(t)} + 2b \left( \langle u_i^{*(t)}, u_i^{*(t)} \rangle_{\mathcal{H}^{(t)}} - 1 \right) \right) c_{ik} = 0$$

which implies that either $c_{ik} = 0$ or $\langle u_i^{*(t)}, u_i^{*(t)} \rangle_{\mathcal{H}^{(t)}} = 1 - \frac{\lambda_{\sigma(k)}^{(t)}}{b}$. Note that $c_{ik}$ and $c_{ij}$ can both be simultaneously non-zero only if $\lambda_{\sigma(k)}^{(t)} = \lambda_{\sigma(j)}^{(t)}$, i.e, $u_i^{*(t)}$ is a linear combination of eigenvectors for the same eigenvalue. Thus, we conclude that either $u_i^{*(t)} = e_{\sigma(i)}^{(t)}$ and $\langle e_{\sigma(i)}^{(t)}, e_{\sigma(i)}^{(t)} \rangle_{\mathcal{H}^{(t)}} = 1 - \frac{\lambda_{\sigma(i)}^{(t)}}{b}$ or $u_i^{*(t)} = 0$. For non-zero, $u_i^{*(t)}$ it is required that $b > \lambda_{\sigma(i)}^{(t)}$. $\square$

### A.2. Proof of Theorem 4.4

*Proof.* Let

$$g^{(t)}(u) = \begin{bmatrix} g_1^{(t)}(u) \\ g_2^{(t)}(u) \\ \vdots \\ g_d^{(t)}(u) \end{bmatrix}, \tag{15}$$

where $g_1^{(t)}$ is defined in (9).

We start by computing the Jacobian of $g^{(t)}$ while applying the stop gradient operator. The matrix $\mathbf{J}^{(t)} = J(g^{(t)})$ is defined such that each row of the matrix corresponds to the gradient of an entry of $g^{(t)}$. We choose to apply the stop gradient operator when computing the Jacobian as optimizing the loss functions with the stop gradient operator is analogous to solving for $u_i$'s sequentially while fixing $u_j$ where $j < i$ as shown by Gomez et al. (2023). Analyzing the stability of those sequential losses would not include a cross gradient term between $u_i$ and $u_j$.

To determine the stability of the equilibrium points, we analyze eigenvalues of the Jacobian evaluated at them (Chicone, 2006). Let $m_i = \mathbb{1}_{u_i^{*(t)} \neq 0}$, and $\rho_{\text{diag}}^{(t)}$ a diagonal matrix where $\rho_{\text{diag}}^{(t)}(s, s) = \rho^{(t)}(s)$ then

$$\mathbf{J}_{ij}^{(t)}(u^{(t)}) = (\nabla_{u_i} g_{u_j}^{(t)}(u)^\top)^\top$$
$$\begin{cases} 2L_{\rho^{(t)}}^{(t)} \odot (\rho^{(t)} \otimes \mathbf{1}) + 2b \left( \langle u_i^{(t)}, u_i^{(t)} \rangle_{\mathcal{H}^{(t)}} - 1 \right) \rho_{\text{diag}}^{(t)} + & \text{, if } i = j \\ \quad 2b \left( 2(u_i^{(t)} \odot \rho^{(t)}) \otimes (u_i^{(t)} \odot \rho^{(t)}) + \sum_{k=1}^{i-1} (u_k^{(t)} \odot \rho^{(t)}) \otimes (u_k^{(t)} \odot \rho^{(t)}) \right) & \\ 0 & \text{, otherwise} \end{cases} \tag{16}$$

Substituting the equilibrium points with the form derived in Lemma 4.3, i.e $u_i^{*(t)} = e_{\sigma(i)}^{(t)} m_i$, $\langle u_i^{(t)}, u_i^{(t)} \rangle_{\mathcal{H}^{(t)}} = $

$\left(1 - \frac{\lambda^{(t)}_{\sigma(i)}}{b}\right) m_i$, and $\langle u^{(t)}_i, u^{(t)}_j \rangle_{\mathcal{H}^{(t)}} = 0$ for $i \neq j$ we get,

$$
\begin{aligned}
\mathbf{J}^{(t)}_{ij}(u^{*(t)}) &= (\nabla_{u_i} g^{(t)}_{u_j}(u^{*(t)}_i)^\top)^\top \\
&= \begin{cases} 
2L^{(t)}_{\rho^{(t)}} \odot (\rho^{(t)} \otimes \mathbf{1}) - 2\lambda^{(t)}_{\sigma(i)} \rho^{(t)}_{\text{diag}} m_i + 2b\rho^{(t)}_{\text{diag}}(m_i - 1)+ & \text{, if } i = j \\
\quad 4b(e^{(t)}_{\sigma(i)} \odot \rho^{(t)}) \otimes (e^{(t)}_{\sigma(i)} \odot \rho^{(t)})m_i+ & \\
\quad 2b\sum_{k=1}^{i-1}(e^{(t)}_{\sigma(k)} \odot \rho^{(t)}) \otimes (e^{(t)}_{\sigma(k)} \odot \rho^{(t)})m_k & \\
0 & \text{, otherwise}
\end{cases}
\end{aligned}
\tag{17}
$$

Note that $\mathbf{J}^{(t)}$ is a block diagonal matrix and its eigenvalues are the union of the diagonal blocks. We proceed to analyze the conditions for the block matrices to be positive definite, i.e when $\langle v_i, \mathbf{J}^{(t)}_{ii}(u^{*(t)})v_i \rangle$ is greater than zero $\forall v_i \in \{v \in \mathbb{R}^{|\mathcal{S}|} : v \neq 0\}$. Since the Laplacian operator is self-adjoint, the eigenvectors form a basis for $\mathbb{R}^{|\mathcal{S}|}$, we can represent each $v_i$ as a linear combination of eigenvectors. Let $v_i = \sum_{k=1}^{|\mathcal{S}|} c_{ik} e^{(t)}_{\sigma(k)}$ in $\langle v_i, \mathbf{J}^{(t)}_{ii}(u^{*(t)})v_i \rangle$ we get $\langle \sum_{k=1}^{|\mathcal{S}|} c_{ik} e^{(t)}_{\sigma(k)}, \mathbf{J}^{(t)}_{ii}(u^{*(t)}) \sum_{k=1}^{|\mathcal{S}|} c_{ik} e^{(t)}_{\sigma(k)} \rangle$.

We first compute $\mathbf{J}^{(t)}_{ii}(u^{*(t)}) \sum_{k=1}^{|\mathcal{S}|} c_k e^{(t)}_{\sigma(k)}$ by replacing $\mathbf{J}_{ii}(u^{*(t)})^{(t)}$ with (17), we get

$$
\begin{aligned}
\mathbf{J}^{(t)}_{ii}&(u^{*(t)}) \sum_{k=1}^{|\mathcal{S}|} c_k e^{(t)}_{\sigma(k)} = \\
&\left(2L^{(t)}_{\rho^{(t)}} \odot (\rho^{(t)} \otimes \mathbf{1}) - 2\lambda^{(t)}_{\sigma(i)} \rho^{(t)}_{\text{diag}} m_i + 2b\rho^{(t)}_{\text{diag}}(m_i - 1)\right) \sum_{k=1}^{|\mathcal{S}|} c_{ik} e^{(t)}_{\sigma(k)}+ \\
&\left(4b(e^{(t)}_{\sigma(i)} \odot \rho^{(t)}) \otimes (e^{(t)}_{\sigma(i)} \odot \rho^{(t)})m_i + 2b\sum_{j=1}^{i-1}(e^{(t)}_{\sigma(j)} \odot \rho^{(t)}) \otimes (e^{(t)}_{\sigma(j)} \odot \rho^{(t)})m_k\right) \sum_{k=1}^{|\mathcal{S}|} c_{ik} e^{(t)}_{\sigma(k)}
\end{aligned}
\tag{18}
$$

Note that

$$
\left((e^{(t)}_{\sigma(j)} \odot \rho^{(t)}) \otimes (e^{(t)}_{\sigma(j)} \odot \rho^{(t)})\right) e^{(t)}_{\sigma(k)} = 0 \; \forall k \neq j
$$

and

$$
\begin{aligned}
\left((e^{(t)}_{\sigma(j)} \odot \rho^{(t)}) \otimes (e^{(t)}_{\sigma(j)} \odot \rho^{(t)})\right) e^{(t)}_{\sigma(j)} &= \langle e^{(t)}_{\sigma(j)}, e^{(t)}_{\sigma(j)} \rangle_{\mathcal{H}^{(t)}} (e^{(t)}_{\sigma(j)} \odot \rho^{(t)}) \\
&= \left(1 - \frac{\lambda^{(t)}_{\sigma(j)}}{b}\right)(e^{(t)}_{\sigma(j)} \odot \rho^{(t)}).
\end{aligned}
$$

Also note that $2L^{(t)}_{\rho^{(t)}} \odot (\rho^{(t)} \otimes \mathbf{1})$ is a matrix with $\left(2L^{(t)}_{\rho^{(t)}} \odot (\rho^{(t)} \otimes \mathbf{1})\right)(s, s') = L(s, s')\rho^{(t)}(s')\rho^{(t)}(s)$, and therefore for any $x \in \mathbb{R}^{|\mathcal{S}|}$

$$
\left(2L^{(t)}_{\rho^{(t)}} \odot (\rho^{(t)} \otimes \mathbf{1})\right) x = 2(Lx) \odot \rho^{(t)}.
\tag{19}
$$

Substituting in (18) we get,

$$
\begin{aligned}
\mathbf{J}^{(t)}_{ii}&(u^{*(t)}) \sum_{k=1}^{|\mathcal{S}|} c_k e^{(t)}_{\sigma(k)} = \\
&\sum_{j=1}^{|\mathcal{S}|} \left(2\left(\lambda^{(t)}_{\sigma(j)} - \lambda^{(t)}_{\sigma(i)} m_i\right) + 2b(m_i - 1)\right) c_{ij}(e^{(t)}_{\sigma(j)} \odot \rho^{(t)}) \\
&+ 4bc_{ii}(e^{(t)}_{\sigma(i)} \odot \rho^{(t)})m_i - 4c_{ii}\lambda^{(t)}_{\sigma(i)}(e^{(t)}_{\sigma(i)} \odot \rho^{(t)})m_i \\
&+ \sum_{j=1}^{i-1} 2bc_{ij}(e^{(t)}_{\sigma(j)} \odot \rho^{(t)})m_j - \sum_{j=1}^{i-1} 2c_{ij}\lambda^{(t)}_{\sigma(j)}(e^{(t)}_{\sigma(j)} \odot \rho^{(t)})m_j
\end{aligned}
\tag{20}
$$

Now we reduced $\mathbf{J}_{ii}^{(t)}(u^{*(t)}) \sum_{k=1}^{|\mathcal{S}|} c_k e_{\sigma(k)}^{(t)}$ to a linear combination $(e_{\sigma(1)}^{(t)} \odot \rho^{(t)}, e_{\sigma(2)}^{(t)} \odot \rho^{(t)}, ..., e_{\sigma(|\mathcal{S}|)}^{(t)} \odot \rho^{(t)})$ with some coefficients $(a_1, a_2, ..., a_{|\mathcal{S}|})$. Since $\langle c_{ij} e_{\sigma(j)}^{(t)}, a_k c_{ik} e_{\sigma(k)}^{(t)} \odot \rho^{(t)} \rangle = a_k c_{ik} c_{ij} \langle e_{\sigma(j)}^{(t)}, e_{\sigma(k)}^{(t)} \rangle_{\mathcal{H}^{(t)}}$ and $\langle e_{\sigma(j)}^{(t)}, e_{\sigma(k)}^{(t)} \rangle_{\mathcal{H}^{(t)}} = 0$ for $j \neq k$ we have

$$\langle \sum_{k=1}^{|\mathcal{S}|} c_{ik} e_{\sigma(k)}^{(t)}, \mathbf{J}_{ii}^{(t)}(u^{*(t)}) \sum_{k=1}^{|\mathcal{S}|} c_{ik} e_{\sigma(k)}^{(t)} \rangle = \sum_{k=1}^{|\mathcal{S}|} a_k c_{ik}^2 \langle e_{\sigma(k)}^{(t)}, e_{\sigma(k)}^{(t)} \rangle_{\mathcal{H}^{(t)}} \tag{21}$$

Since $\langle e_{\sigma(k)}^{(t)}, e_{\sigma(k)}^{(t)} \rangle_{\mathcal{H}^{(t)}} > 0$ and $c_{ik}^2 \geq 0$, $a_k$ must be positive $\forall k$ for $\mathbf{J}_{ii}^{(t)}(u^{*(t)})$ to be positive definite. We group the conditions from (20) that are required to be positive below

$$\begin{cases} 2b(m_i + m_j - 1) - 2\lambda_{\sigma(i)}^{(t)} m_i + 2\lambda_{\sigma(j)}^{(t)}(1 - m_j) & \forall 1 \leq j < i \leq d \\ 6bm_i + 2\lambda_{\sigma(i)}^{(t)} - 6\lambda_{\sigma(i)}^{(t)} m_i - 2b & \forall 1 \leq i \leq d \\ 2(\lambda_{\sigma(j)}^{(t)} - \lambda_{\sigma(i)}^{(t)} m_i) + 2b(m_i - 1) & \forall 1 \leq i < j \leq |\mathcal{S}|. \end{cases} \tag{22}$$

If any $u_i^{*(t)} = 0$, then the third condition becomes $2\lambda_{\sigma(j)}^{(t)} - 2b$ which is always negative under the selection of hyperparameters discussed in Lemma 4.3, hence it is unstable. For equilibrium points where all $u_i^{*(t)}$ are non-zero, i.e $m_i = 1 \forall i$, the conditions becomes

$$\begin{cases} 2b - 2\lambda_{\sigma(i)}^{(t)} & \forall 1 \leq j < i \leq d \\ 4b - 4\lambda_{\sigma(i)}^{(t)} & \forall 1 \leq i \leq d \\ 2(\lambda_{\sigma(j)}^{(t)} - \lambda_{\sigma(i)}^{(t)}) & \forall 1 \leq i < j \leq |\mathcal{S}|. \end{cases} \tag{23}$$

The third condition indicates that $2(\lambda_{\sigma(j)}^{(t)} - \lambda_{\sigma(i)}^{(t)})$ has to be positive which is only true for the identity permutation and if the maximum eigenvalue multiplicity of the Laplacian is 1. The second and first conditions imply that $b - \lambda_{\sigma(i)}^{(t)}$ must be positive which is true when $b > \lambda_{\sigma(i)}^{(t)} \forall 1 \leq i \leq |\mathcal{S}|$ which is already a requirement of Lemma 4.3. □

## A.3. Proof of Proposition 4.5

*Proof.* To show that the gradient function $g^{(t)}$ is Lipschitz continuous, we proceed to show that the spectral norm of the Jacobian is bounded $\forall u \in \mathcal{C}^{(t)}$. Notice that the Jacobian defined in (16) is a block diagonal matrix, hence its singular values are the union of the singular values of the block matrices $\mathbf{J}_{ii}^{(t)}(u)$, and $\|\mathbf{J}^{(t)}(u)\| = \max_i \|\mathbf{J}_{ii}^{(t)}(u)\|$. By the triangle inequality we have,

$$\|\mathbf{J}_{ii}^{(t)}(u)\| \leq \left\| 2L_{\rho^{(t)}}^{(t)} \odot (\rho^{(t)} \otimes \mathbf{1}) \right\| + \left\| 2b \left( \langle u_i^{(t)}, u_i^{(t)} \rangle_{\mathcal{H}^{(t)}} - 1 \right) \rho_{\text{diag}}^{(t)} \right\| + \tag{24}$$

$$\left\| 4b(u_i^{(t)} \odot \rho^{(t)}) \otimes (u_i^{(t)} \odot \rho^{(t)}) \right\| + 2b \sum_{k=1}^{i-1} \left\| (u_k^{(t)} \odot \rho^{(t)}) \otimes (u_k^{(t)} \odot \rho^{(t)}) \right\|$$

We start by bounding the first term, by (19) we know that for any vector $x \in \mathbb{R}^{|\mathcal{S}|}$, $\left( 2L_{\rho^{(t)}}^{(t)} \odot (\rho^{(t)} \otimes \mathbf{1}) \right) x = 2(Lx) \odot \rho^{(t)}$. For any $x \in \mathbb{R}^{|\mathcal{S}|}$ with $\|x\| = 1$,

$$\left\| \left( L_{\rho^{(t)}}^{(t)} \odot (\rho^{(t)} \otimes \mathbf{1}) \right) x \right\| = \left\| (Lx) \odot \rho^{(t)} \right\| = \sqrt{\langle (Lx) \odot \rho^{(t)}, (Lx) \odot \rho^{(t)} \rangle}$$

$$= \sqrt{\sum_{s \in \mathcal{S}} ((Lx)(s))^2 \rho^{(t)}(s)^2} \leq \sqrt{\sum_{s \in \mathcal{S}} ((Lx)(s))^2 \rho^{(t)}(s)}$$

$$= \sqrt{\langle (Lx), (Lx) \rangle_{\mathcal{H}^{(t)}}} = \|Lx\|_{\mathcal{H}^{(t)}} \leq \|L\|_{\mathcal{H}^{(t)}} \|x\|_{\mathcal{H}^{(t)}} \leq \|L\|_{\mathcal{H}^{(t)}}.$$

Therefore,

$$\left\|\left(2L^{(t)}_{\rho^{(t)}} \odot (\rho^{(t)} \otimes \mathbf{1})\right)\right\| \leq 2\|L\|_{\mathcal{H}^{(t)}} \overset{(i)}{\leq} 2, \tag{25}$$

where $(i)$ follows from $\|L\|_{\mathcal{H}^{(t)}} \leq 1$ (Wu et al., 2018).

For the second term, since $\|\rho^{(t)}_{\text{diag}}\| \leq 1$ and $\langle u^{(t)}_i, u^{(t)}_i \rangle_{\mathcal{H}^{(t)}} \leq \frac{2}{\rho_{min}}$, we have

$$\left\|2b\left(\langle u^{(t)}_i, u^{(t)}_i \rangle_{\mathcal{H}^{(t)}} - 1\right)\rho^{(t)}_{\text{diag}}\right\| \leq 2b\left\|\langle u^{(t)}_i, u^{(t)}_i \rangle_{\mathcal{H}^{(t)}}\rho^{(t)}_{\text{diag}}\right\| + 2b\left\|\rho^{(t)}_{\text{diag}}\right\| \leq \frac{4b}{\rho_{min}} + 2b. \tag{26}$$

For the remaining terms, note that for any $x \in \mathbb{R}^{|\mathcal{S}|}$ with $\|x\| = 1$,

$$\left\|\left((u^{(t)}_i \odot \rho^{(t)}) \otimes (u^{(t)}_i \odot \rho^{(t)})\right)x\right\| = \left\|\langle u^{(t)}_i, x \rangle_{\mathcal{H}^{(t)}}(u^{(t)}_i \odot \rho^{(t)})\right\| \leq \|x\|_{\mathcal{H}^{(t)}}\|u^{(t)}_i\|_{\mathcal{H}^{(t)}}\left\|u^{(t)}_i \odot \rho^{(t)}\right\|$$

$$\leq \|x\|\|u^{(t)}_i\|\left\|u^{(t)}_i \odot \mathbf{1}\right\| = \|u^{(t)}_i\|^2 \overset{(i)}{\leq} \frac{2}{\rho_{min}}, \tag{27}$$

where $(i)$ follows from $u^{(t)}_i$ being an element of the constraint set $\mathcal{C}^{(t)}$.

Combining equations 25, 26, and 27 in (24), we get

$$\|\mathbf{J}^{(t)}_{ii}(u)\| \leq \alpha = 2 + \left(2 + \frac{12 + 4d}{\rho_{min}}\right)b.$$

As the spectral norm of each block matrix $J^{(t)}_{ii}(u)$ is bounded by $\alpha$ $\forall u \in \mathcal{C}^{(t)}$, the spectral norm of $J^{(t)}(u)$ is bounded by $\alpha$ and the gradient $g^{(t)}$ is Lipschitz continuous with the Lipschitz constant $\alpha$ $\forall u \in \mathcal{C}^{(t)}$. $\qquad\square$

### A.4. Proof of Lemma 4.6

**Proof for Lemma 4.6 (a)**

*Proof.* We denote $A(i, :)$ as the $i$-th row of the matrix $A$.

$$\left\|P^{(t+1)} - P^{(t)}\right\|_{\infty} = \max_{s \in \mathcal{S}}\left\|P^{(t+1)}(s, :) - P^{(t)}(s, :)\right\|_1$$

$$= \max_{s \in \mathcal{S}}\left\|\sum_{a \in \mathcal{A}}(\pi^{(t+1)}(a|s) - \pi^{(t)}(a|s))\mathcal{T}(s, a, :)\right\|_1$$

$$\overset{(i)}{\leq} \max_{s \in \mathcal{S}}\sum_{a \in \mathcal{A}}\left|\pi^{(t+1)}(a|s) - \pi^{(t)}(a|s)\right|\|\mathcal{T}(s, a, :)\|_1$$

$$\overset{(ii)}{=} \max_{s \in \mathcal{S}}\sum_{a \in \mathcal{A}}\left|\pi^{(t+1)}(a|s) - \pi^{(t)}(a|s)\right| = \delta^{(t)}_{\pi},$$

where (i) is by the triangle inequality, and (ii) from the fact that $\|\mathcal{T}(s, a, :)\|_1 = 1$. $\qquad\square$

**Proof for Lemma 4.6 (c)**

*Proof.* First note that the elements of the matrix $(\rho^{(t)} \otimes \mathbf{1}) \odot L^{(t)}_{\rho^{(t)}}$ are defined as

$$(\rho^{(t)} \otimes \mathbf{1}) \odot L^{(t)}_{\rho^{(t)}}(s, s') = \rho^{(t)}(s)\mathbb{1}_{s=s'} - \rho^{(t)}(s)W^{(t)}(s, s')\rho^{(t)}(s')$$

$$= \rho^{(t)}(s)\mathbb{1}_{s=s'} - \frac{1}{2}P^{(t)}(s, s')\rho^{(t)}(s) - \frac{1}{2}P^{(t)}(s', s)\rho^{(t)}(s').$$

Hence, by applying the triangle inequality, we have

$$\left\| (\rho^{(t+1)} \otimes \mathbf{1}) \odot L^{(t+1)}_{\rho^{(t+1)}} - (\rho^{(t)} \otimes \mathbf{1}) \odot L^{(t)}_{\rho^{(t)}} \right\|$$

$$= \left\| \rho^{(t+1)}_{\text{diag}} - \rho^{(t)}_{\text{diag}} - (\rho^{(t+1)} \otimes \mathbf{1}) \odot W^{(t+1)}_{\rho^{(t+1)}} + (\rho^{(t)} \otimes \mathbf{1}) \odot W^{(t)}_{\rho^{(t)}} \right\|$$

$$\leq \left\| \rho^{(t+1)}_{\text{diag}} - \rho^{(t)}_{\text{diag}} \right\| + \left\| (\rho^{(t+1)} \otimes \mathbf{1}) \odot W^{(t+1)}_{\rho^{(t+1)}} - (\rho^{(t)} \otimes \mathbf{1}) \odot W^{(t)}_{\rho^{(t)}} \right\|$$

$$\leq \delta^{(t)}_\rho + \left\| (\rho^{(t+1)} \otimes \mathbf{1}) \odot W^{(t+1)}_{\rho^{(t+1)}} - (\rho^{(t)} \otimes \mathbf{1}) \odot W^{(t)}_{\rho^{(t)}} \right\|$$

$$\leq \delta^{(t)}_\rho + \frac{1}{2} \left\| (\rho^{(t+1)} \otimes \mathbf{1}) \odot P^{(t+1)} - (\rho^{(t)} \otimes \mathbf{1}) \odot P^{(t)} \right\|$$

$$+ \frac{1}{2} \left\| \left( (\rho^{(t+1)} \otimes \mathbf{1}) \odot P^{(t+1)} - (\rho^{(t)} \otimes \mathbf{1}) \odot P^{(t)} \right)^\top \right\|$$

And since $\|A^\top\| = \|A\|$ we have

$$\left\| (\rho^{(t+1)} \otimes \mathbf{1}) \odot L^{(t+1)}_{\rho^{(t+1)}} - (\rho^{(t)} \otimes \mathbf{1}) \odot L^{(t)}_{\rho^{(t)}} \right\| \tag{28}$$
$$\leq \delta^{(t)}_\rho + \left\| (\rho^{(t+1)} \otimes \mathbf{1}) \odot P^{(t+1)} - (\rho^{(t)} \otimes \mathbf{1}) \odot P^{(t)} \right\|$$

Now we proceed to bound the second term, adding and subtracting $(\rho^{(t+1)} \otimes \mathbf{1}) \odot P^{(t)}$ and applying the triangle inequality we have

$$\left\| (\rho^{(t+1)} \otimes \mathbf{1}) \odot P^{(t+1)} - (\rho^{(t)} \otimes \mathbf{1}) \odot P^{(t)} \right\|$$

$$\leq \left\| (\rho^{(t+1)} \otimes \mathbf{1}) \odot (P^{(t+1)} - P^{(t)}) \right\| + \left\| ((\rho^{(t+1)} \otimes \mathbf{1}) - \rho^{(t)} \otimes \mathbf{1})) \odot P^{(t)} \right\|$$

$$\overset{(i)}{\leq} \sqrt{|\mathcal{S}|} \max_{s \in \mathcal{S}} \left\| \rho^{(t+1)}(s) \left( P^{(t+1)}(s,:) - P^{(t)}(s,:) \right) \right\|_1$$

$$+ \sqrt{|\mathcal{S}|} \max_{s \in \mathcal{S}} \left\| \left( \rho^{(t+1)}(s) - \rho^{(t)}(s) \right) P^{(t)}(s,:) \right\|_1$$

$$\leq \sqrt{|\mathcal{S}|} \left\| \rho^{(t+1)} \right\|_\infty \max_{s \in \mathcal{S}} \left\| P^{(t+1)}(s,:) - P^{(t)}(s,:) \right\|_1$$

$$+ \sqrt{|\mathcal{S}|} \left\| \rho^{(t+1)} - \rho^{(t)} \right\|_\infty \max_{s \in \mathcal{S}} \left\| P^{(t)}(s,:) \right\|_1$$

$$\overset{(ii)}{\leq} \sqrt{|\mathcal{S}|} \left( \delta^{(t)}_\pi + \delta^{(t)}_\rho \right)$$

where $(i)$ stems from the identity $\|A\| \leq \sqrt{n}\|A\|_\infty$ for the $n \times n$ matrix $A$ and $(ii)$ follows from $\left\| \rho^{(t+1)} \right\|_\infty \leq 1$, $\left\| P^{(t)}(s,:) \right\|_1 = 1$, and Lemma 4.6(a). $\qquad \square$

**Proof for Lemma 4.6 (d)**

*Proof.* Recall that the loss function is given by:

$$\mathcal{L}^{(t)}(u) = \sum_{i=1}^d \langle u_i, L^{(t)} u_i \rangle_{\mathcal{H}^{(t)}} + b \sum_{j=1}^d \sum_{k=1}^{j-1} (\langle u_j, [\![u_k]\!] \rangle_{\mathcal{H}^{(t)}})^2 + \frac{b}{2} \sum_{i=1}^d (\langle u_i, u_i \rangle_{\mathcal{H}^{(t)}} - 1)^2 \tag{29}$$

We are interested in finding a bound for the difference:

$$\Delta \mathcal{L}^{(t)}(u) = |\mathcal{L}^{(t+1)}(u) - \mathcal{L}^{(t)}(u)|. \tag{30}$$

The first term in the loss function is:

$$\sum_{i=1}^d \langle u_i, L^{(t)} u_i \rangle_{\mathcal{H}^{(t)}}. \tag{31}$$

Substituting the inner product and applying the triangle inequality, we have the following:

$$
\left| \sum_{i=1}^{d} \langle u_i, L^{(t+1)} u_i \rangle_{\mathcal{H}^{(t+1)}} - \sum_{i=1}^{d} \langle u_i, L^{(t)} u_i \rangle_{\mathcal{H}^{(t)}} \right| \leq
$$
$$
\sum_{i=1}^{d} \left| \left( (u_i^\top \odot \rho^{(t+1)^\top}) L_{\rho^{(t+1)}}^{(t+1)}(u_i) - (u_i^\top \odot \rho^{(t)^\top}) L_{\rho^{(t)}}^{(t)}(u_i) \right) \right|. \tag{32}
$$

The above expression can be re-written as follows:

$$
\sum_{i=1}^{d} \left| \left( u_i^\top \left( (\rho^{(t+1)} \otimes \mathbf{1}) \odot L_{\rho^{(t+1)}}^{(t+1)} - (\rho^{(t)} \otimes \mathbf{1}) \odot L_{\rho^{(t)}}^{(t)} \right) u_i \right) \right|. \tag{33}
$$

From Lemma 4.6(c), $\left\| (\rho^{(t+1)} \otimes \mathbf{1}) \odot L_{\rho^{(t+1)}}^{(t+1)} - (\rho^{(t)} \otimes \mathbf{1}) \odot L_{\rho^{(t)}}^{(t)} \right\| \leq \delta_L^{(t)}$. Thus, we have:

$$
\sum_{i=1}^{d} \left| \left( u_i^\top \left( (\rho^{(t+1)} \otimes \mathbf{1}) \odot L_{\rho^{(t+1)}}^{(t+1)} - (\rho^{(t)} \otimes \mathbf{1}) \odot L_{\rho^{(t)}}^{(t)} \right) u_i \right) \right| \leq \delta_L^{(t)} \sum_{i=1}^{d} \|u_i\|^2 \tag{34}
$$

The difference in the regularization terms is:

$$
\left| b \sum_{j=1}^{d} \sum_{k=1}^{j-1} \left( (\langle u_j, [\![u_k]\!] \rangle_{\mathcal{H}^{(t+1)}})^2 - (\langle u_j, [\![u_k]\!] \rangle_{\mathcal{H}^{(t)}})^2 \right) + \right.
$$
$$
\left. \frac{b}{2} \sum_{j=1}^{d} \left( (\langle u_j, u_j \rangle_{\mathcal{H}^{(t+1)}} - 1)^2 - (\langle u_j, u_j \rangle_{\mathcal{H}^{(t)}} - 1)^2 \right) \right|. \tag{35}
$$

Using the rule $x^2 - y^2 = (x+y) \cdot (x-y)$ and applying the triangle inequality, we can rewrite the above expression as follows:

$$
b \sum_{j=1}^{d} \sum_{k=1}^{j-1} |\langle u_j, [\![u_k]\!] \rangle_{\mathcal{H}^{(t+1)}} + \langle u_j, [\![u_k]\!] \rangle_{\mathcal{H}^{(t)}}| \, |\langle u_j, [\![u_k]\!] \rangle_{\mathcal{H}^{(t+1)}} - \langle u_j, [\![u_k]\!] \rangle_{\mathcal{H}^{(t)}}| +
$$
$$
\frac{b}{2} \sum_{j=1}^{d} |\langle u_j, u_j \rangle_{\mathcal{H}^{(t+1)}} + \langle u_j, u_j \rangle_{\mathcal{H}^{(t)}} - 2| \, |(\langle u_j, u_j \rangle_{\mathcal{H}^{(t+1)}} - \langle u_j, u_j \rangle_{\mathcal{H}^{(t)}})|
$$
$$
\leq b \sum_{j=1}^{d} \sum_{k=1}^{j-1} |\langle u_j, [\![u_k]\!] \rangle_{\mathcal{H}^{(t+1)}} + \langle u_j, [\![u_k]\!] \rangle_{\mathcal{H}^{(t)}}| \, |(\langle u_j, [\![u_k]\!] \rangle_{\mathcal{H}^{(t+1)}} - \langle u_j, [\![u_k]\!] \rangle_{\mathcal{H}^{(t)}})| +
$$
$$
\frac{b}{2} \sum_{j=1}^{d} |\langle u_j, u_j \rangle_{\mathcal{H}^{(t+1)}} + \langle u_j, u_j \rangle_{\mathcal{H}^{(t)}}| \, |(\langle u_j, u_j \rangle_{\mathcal{H}^{(t+1)}} - \langle u_j, u_j \rangle_{\mathcal{H}^{(t)}})| \tag{36}
$$

Note that

$$
|\langle u_j, [\![u_k]\!] \rangle_{\mathcal{H}^{(t+1)}} + \langle u_j, [\![u_k]\!] \rangle_{\mathcal{H}^{(t)}}| \leq 2\|u_j\| \cdot \|[\![u_k]\!]\| \tag{37}
$$

and that

$$
|\langle u_j, [\![u_k]\!] \rangle_{\mathcal{H}^{(t+1)}} - \langle u_j, [\![u_k]\!] \rangle_{\mathcal{H}^{(t)}}| = \left| \sum_{s \in \mathcal{S}} u_j(s)(\rho^{(t+1)}(s) - \rho^{(t)}(s))[\![u_k]\!](s) \right|
$$
$$
\leq \|u_j\| \cdot \|[\![u_k]\!]\| \cdot \|\rho^{(t+1)} - \rho^{(t)}\|_\infty \leq \|u_j\| \cdot \|[\![u_k]\!]\| \cdot \delta_\rho^{(t)} \tag{38}
$$

where $\delta_\rho^{(t)}$ is defined in Lemma 4.6(b).

Combining the bounds for both the first and second parts, the total bound on $\mathcal{L}^{(t+1)}(u) - \mathcal{L}^{(t)}(u)$ is:

$$|\mathcal{L}^{(t+1)}(u) - \mathcal{L}^{(t)}(u)| \leq \delta_L^{(t)} \sum_{i=1}^{d} \|u_i\|^2 + b \sum_{j=1}^{d} \sum_{k=1}^{j} \left( 2\|u_j\|^2 \|[\![u_k]\!]\|^2 \delta_\rho^{(t)} \right). \tag{39}$$

We have $|\mathcal{L}^{(t+1)}(u) - \mathcal{L}^{(t)}(u)| \leq \delta_{\mathcal{L}}^{(t)}$, where $\delta_{\mathcal{L}}^{(t)}$ is given by

$$\delta_{\mathcal{L}}^{(t)} = \delta_L^{(t)} \sum_{i=1}^{d} \|u_i\|^2 + b \sum_{j=1}^{d} \sum_{k=1}^{j} \left( 2\|u_j\|^2 \|[\![u_k]\!]\|^2 \delta_\rho^{(t)} \right). \tag{40}$$

We know that $\|u_i\|^2 \leq 2/\rho_{min}$. Substituting this, we have

$$\delta_{\mathcal{L}}^{(t)} = \frac{2d\delta_L^{(t)}}{\rho_{min}} + \frac{8bd^2\delta_\rho^{(t)}}{\rho_{min}^2}. \tag{41}$$

Note: From Lemma 4.6(b) and Lemma 4.6(c), we have $\delta_L^{(t)} \leq C_1 \delta_\pi^{(t)}$ and $\delta_\rho^{(t)} \leq C_2 \delta_\pi^{(t)}$, for some constants $C_1, C_2$. Thus, we have $|\mathcal{L}^{(t+1)}(u) - \mathcal{L}^{(t)}(u)| \leq \delta_{\mathcal{L}}^{(t)} = (C_1 + C_2)\delta_\pi^{(t)}$. This implies that the drift in the loss function decreases with the decrease in the drift between the policies $\pi_t$ and $\pi_{t+1}$.

$\square$

### A.5. Proof of Theorem 4.7

*Proof.* Recall that the update rule for projected gradient descent in (10) is given by:

$$u_i^{(t+1)} \leftarrow u_i^{(t)} - \eta G_{u_i}^{(t)}(u_i^{(t)}),$$

We need to prove that the gradient norm $\|g^{(t)}(u_t)\|$ asymptotically approaches zero as $t \to \infty$, which would ensure the convergence to a critical point. In order to prove this, we will establish that the sum of the squared gradients remains finite over time, despite the loss function being time-varying.

Recall the following assumptions:

- The gradient of the time-varying loss function $\mathcal{L}^{(t)}(u)$ is Lipschitz continuous with constant $\alpha > 0$ for all $t$, that is,

$$\|g^{(t)}(u_1) - \nabla_u g^{(t)}(u_2)\| \leq \alpha \|u_1 - u_2\|, \quad \forall u_1, u_2.$$

- From Lemma 4.6, we have the change in the loss function from time $t$ to time $t + 1$ is bounded by a constant $\delta_{\mathcal{L}}$, i.e.,

$$\|\mathcal{L}^{(t+1)}(u) - \mathcal{L}^{(t)}(u)\| \leq \delta_{\mathcal{L}}^{(t)}, \quad \forall u.$$

- Additionally, it is easy to see that the loss function $\mathcal{L}^{(t)}(u)$ is bounded from below by a constant $\mathcal{L}^*$, i.e.,

$$\mathcal{L}^{(t)}(u) \geq \mathcal{L}^*, \quad \forall u, t.$$

The descent lemma for a loss function with Lipschitz continuous gradients and learning rate $\eta$ is given by:

$$\mathcal{L}^{(t+1)}(u^{(t+1)}) \leq \mathcal{L}^{(t+1)}(u^{(t)}) - \eta \|G^{(t)}(u^{(t)})\|^2 + \frac{\eta^2}{2}\alpha \|G^{(t)}(u^{(t)})\|^2.$$

This can be rewritten as:

$$\mathcal{L}^{(t+1)}(u^{(t+1)}) \leq \mathcal{L}^{(t)}(u^{(t)}) - \eta \|G^{(t)}(u^{(t)})\|^2 + \frac{\eta^2}{2}\alpha \|G^{(t)}(u^{(t)})\|^2 + \delta_{\mathcal{L}}^{(t)},$$

where $\delta_{\mathcal{L}}^{(t)}$ represents the drift that accounts for the time-variation in the loss function between time $t$ and $t+1$. Rearranging this inequality, we obtain:

$$\mathcal{L}^{(t+1)}(u^{(t+1)}) \leq \mathcal{L}^{(t)}(u^{(t)}) - \left(\eta - \frac{\eta^2}{2}\alpha\right)\|G^{(t)}(u^{(t)})\|^2 + \delta_{\mathcal{L}}^{(t)}.$$

To ensure that the loss function decreases at each time step, except for the small drift $\delta_{\mathcal{L}}$, we require that:

$$\eta - \frac{\eta^2}{2}\alpha > 0.$$

This gives the condition on the learning rate:

$$\eta < \frac{2}{\alpha}.$$

Thus, the learning rate must satisfy $\eta \leq \frac{2}{\alpha}$.

At each step, we can bound the change in the loss function as follows:

$$\mathcal{L}^{(t)}(u^{(t)}) - \mathcal{L}^{(t+1)}(u^{(t+1)}) \geq \left(\eta - \frac{\eta^2}{2}\alpha\right)\|G^{(t)}(u^{(t)})\|^2 - \delta_{\mathcal{L}}^{(t)}.$$

Summing this inequality over $t = 1, 2, \ldots, T$, we get:

$$\sum_{t=1}^{T}\left(\mathcal{L}^{(t)}(u^{(t)}) - \mathcal{L}^{(t+1)}(u^{(t+1)})\right) \geq \sum_{t=1}^{T}\left(\left(\eta - \frac{\eta^2}{2}\alpha\right)\|G^{(t)}(u^{(t)})\|^2 - \delta_{\mathcal{L}}^{(t)}\right).$$

The left-hand side of this inequality is a telescoping sum, so it simplifies to:

$$\mathcal{L}^{(1)}(u^{(1)}) - \mathcal{L}^{(T+1)}(u^{(T+1)}) \geq \sum_{t=1}^{T}\left(\left(\eta - \frac{\eta^2}{2}\alpha\right)\|G^{(t)}(u^{(t)})\|^2 - \delta_{\mathcal{L}}^{(t)}\right).$$

Rearranging, we get:

$$\sum_{t=1}^{T}\|G^{(t)}(u^{(t)})\|^2 \leq \frac{\mathcal{L}^{(1)}(u^{(1)}) - \mathcal{L}^{(T+1)}(u^{(T+1)})}{\eta - \frac{\eta^2}{2}\alpha} + \frac{\sum_{t=1}^{T}\delta_{\mathcal{L}}^{(t)}}{\eta - \frac{\eta^2}{2}\alpha}.$$

Since the loss function $\mathcal{L}^{(t)}(u)$ is bounded from below by $\mathcal{L}^*$, we have:

$$\mathcal{L}^{(1)}(u^{(1)}) - \mathcal{L}^* \geq \sum_{t=1}^{T}\left(\left(\eta - \frac{\eta^2}{2}\alpha\right)\|G^{(t)}(u^{(t)})\|^2 - \delta_{\mathcal{L}}^{(t)}\right).$$

We can further simplify this to:

$$\sum_{t=1}^{T}\|G^{(t)}(u^{(t)})\|^2 \leq \frac{\mathcal{L}^{(1)}(u^{(1)}) - \mathcal{L}^*}{\eta - \frac{\eta^2}{2}\alpha} + \frac{\sum_{t=1}^{T}\delta_{\mathcal{L}}^{(t)}}{\eta - \frac{\eta^2}{2}\alpha}. \tag{42}$$

Dividing both sides by $T$, we get

$$\mathbb{E}_{t \sim \text{Uniform}\{1,2,\ldots,T\}}\left[\|G^{(t)}(u^{(t)})\|^2\right] \leq \frac{\mathcal{L}^{(1)}(u^{(1)}) - \mathcal{L}^*}{T\left(\eta - \frac{\eta^2}{2}\alpha\right)} + \frac{\sum_{t=1}^{T}\delta_{\mathcal{L}}^{(t)}}{T\left(\eta - \frac{\eta^2}{2}\alpha\right)}. \tag{43}$$

Setting $\eta = \frac{1}{\alpha}$, we have

$$\mathbb{E}_{t \sim \text{Uniform}\{1,2,\ldots,T\}}\left[\|G^{(t)}(u^{(t)})\|^2\right] \leq \frac{2\alpha}{T}\left(\mathcal{L}^{(1)}(u^{(1)}) - \mathcal{L}^* + \sum_{t=1}^{T}\delta_{\mathcal{L}}^{(t)}\right). \tag{44}$$

From Assumption 4.2, we have that the asymptotic sum of the squared gradients $\lim_{T\to\infty}\sum_{t=1}^{\infty}\|G^{(t)}(u^{(t)})\|^2$ remains finite, i.e.,

$\lim_{T\to\infty}\sum_{t=1}^{T}\|G^{(t)}(u^{(t)})\|^2 < \infty$. Therefore, we have:

$$\lim_{t\to\infty}\|G^{(t)}(u^{(t)})\| = 0.$$

This shows that the gradients asymptotically approach zero over time, proving that the projected gradient descent algorithm applied to the time-varying loss function converges asymptotically to a critical point.

$\square$

## B. Additional Experimental Details

We provide hyper-parameters for the Asymmetric Graph Drawing Objective (AGDO), Proximal Policy Optimization (PPO), and Deep-Q Network (DQN) in Table 1.

Table 1: Hyper-parameters for AGDO, PPO, and DQN.

| Hyper-Parameter | AGDO | PPO | DQN |
|---|---|---|---|
| $d$ | 11 | - | - |
| Replay Max Episodes | 20 | - | - |
| Updates per Episodic Step | 5 | - | - |
| Total Training Steps | 200,000 | - | - |
| Maximum Episode Length | 10,000 | - | - |
| Learning Rate | 0.001 | $3 \times 10^{-4}$ | $3 \times 10^{-4}$ |
| Optimizer | Adam | Adam | Adam |
| Barrier Coefficient | 5 | - | - |
| Encoder Network Hidden Dimensions | [256, 256, 256] | - | - |
| Batch Size | 256 | 256 | 256 |
| Replay Buffer Size | - | 500 steps | 50,000 steps |
| Update Every | - | 500 steps | 1 step |
| Training Batches per Update | - | 10 | 1 |
| Actor and Critic Hidden Dimensions | - | [64, 64] | - |
| Q-Network Hidden Dimensions | - | - | [64, 64] |
| Discount Factor | - | 0.99 | 0.99 |
| Entropy Coefficient | - | 0.01 | - |
| Initial Clip Ratio | - | 0.2 | - |
| Final Clip Ratio | - | 0.01 | - |
| Initial Epsilon | - | - | 1 |
| Final Epsilon | - | - | 0.1 |

Finally, we present the rewards obtained by the learning agents described in Section 5, as illustrated in Figure 6. Figure 6a shows the policies trained using the online version of AGDO across four different environments. In environments with larger state spaces, the total rewards are comparatively lower.

Figure 5a demonstrates that larger clipping improves the learning representation. However, as discussed in Section 5, this improvement comes at the cost of reduced policy quality and can be seen in Figure 6b.

Figures 6c and 6d depict the effects of tuning parameters that influence only the encoder training. As a result, the quality of the policies remains consistent across these experiments. In Figure 6c, the number of steps per policy update determines how frequently the encoder is updated. In Figure 6d, the size of the replay buffer is varied only for encoder training, while it remains fixed during policy training.

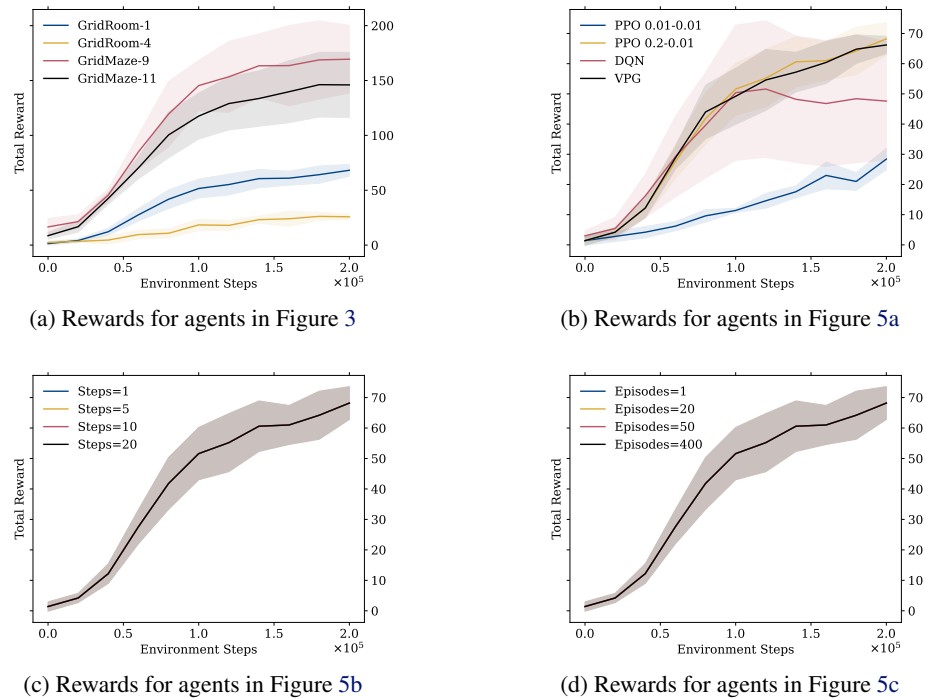

(a) Rewards for agents in Figure 3

(b) Rewards for agents in Figure 5a

(c) Rewards for agents in Figure 5b

(d) Rewards for agents in Figure 5c

Figure 6: Average reward obtained by agents trained in section 5.

