# OpenReview forum: "Online Laplacian-Based Representation Learning in Reinforcement Learning"
_ICML.cc/2025/Conference — ICML 2025 poster_

### Official Review · Reviewer_bt1X · 2025-03-10

**Overall Recommendation:** 2

**Summary:**

This paper studies how to learn a Laplacian-based state representation in an **online** manner for reinforcement learning, rather than fixing a policy and then extracting a representation from its induced Markov chain. The authors propose a modified “asymmetric graph drawing” objective (AGDO) that can be jointly optimized with the policy’s updates, ensuring that the representation automatically adapts to policy changes. They prove that under mild assumptions about how quickly the policy changes, online projected gradient descent on AGDO converges and recovers the true Laplacian embedding of the evolving Markov chain. Empirical experiments in grid-world environments validate these guarantees and show that AGDO-based online learning can match the quality of prior offline Laplacian objectives, with advantages when the policy is updated throughout training.

**Claims And Evidence:**

Yes

**Essential References Not Discussed:**

No

**Experimental Designs Or Analyses:**

Yes, no issue

**Methods And Evaluation Criteria:**

Yes

**Other Comments Or Suggestions:**

please refer to weaknesses

**Other Strengths And Weaknesses:**

## Strength
- The paper makes a clear, novel contribution by theoretically extending Laplacian-based representation learning to time-varying RL settings and confirming convergence under bounded policy drift. This is a welcome step in bridging purely offline methods and practical “end-to-end” RL pipelines that adapt representations as policies improve. The significance lies in demonstrating that Laplacian embeddings can remain valid in a continuously evolving MDP, which in turn may improve exploration and stability in practice.

- One notable strength is the careful balance of theoretical rigor and empirical demonstration, where each assumption (especially the drift bound) is mapped back to commonly used techniques like trust-region updates. The authors also provide experiments confirming that performance benefits hold in multiple grid-world variants and that, indeed, the policy’s drift rate is a key factor. Such clarity on what matters for “online Laplacian learning” makes the result more interpretable and actionable.

## Weakness
- A potential weakness is that real-world RL scenarios could see more drastic or erratic policy shifts, at least early on, than the sublinear bound presumes. The paper does address this by suggesting smaller step sizes or stable policy updates, but more thorough discussion of how to handle genuinely large, unpredictable changes would strengthen the practical impact. Another small drawback is that the experiments revolve around relatively small navigation tasks; whether the same approach is straightforward to integrate in large-scale function approximators (beyond MLPs over low-dimensional input) or more complex continuous-control benchmarks remains open. Nonetheless, the clarity and step-by-step build-up of the main algorithm, plus the thorough theoretical arguments, make the paper a meaningful contribution in bridging offline eigenvector-based representations and real-time RL learning.

- It remains doubt that how much the theoretical contribution of this work relies on the existing work of Gomez et al. (2023). The authors may need to explicitly mention the uniqueness of their theoretical contribution in the main paper.

**Questions For Authors:**

- Q1: he theoretical guarantees assume bounded policy drift. Have you tested cases where the policy undergoes rapid, large updates (e.g., early in training when exploration is high)? Would the method still be stable, or do you observe degradation in representation accuracy?

- Q2: Your experiments focus on grid-world settings with relatively simple representations. Have you explored how well this approach scales to high-dimensional, visual, or continuous-control environments? Would convolutional or attention-based architectures be needed to maintain stability in more complex tasks?

**Relation To Broader Scientific Literature:**

This work extends the established line of **Laplacian-based representation learning**—often called proto-value functions or eigenoptions in reinforcement learning—from an offline, fixed-policy setting to a fully **online** one where the policy (and thus the Markov chain) evolves over training. Prior studies had shown how to approximate Laplacian eigenvectors efficiently or how to break the rotational symmetry in eigenvector solutions, but only for a single static operator. Recent empirical work (for instance, in learning options with deep Laplacian representations) explored updating the representation while the policy changed, but without providing rigorous convergence guarantees.

In that sense, the paper (1) **proposes a variant** of previously known objectives (such as the “augmented Lagrangian Laplacian objective,” ALLO) that uses an “asymmetric” construction and (2) **proves** that projected gradient descent still converges in the presence of modest policy drift. It thus unifies two strands of the literature:
- The original demonstration that one can recover Laplacian eigenvectors through unconstrained objectives, plus symmetrical-breaking terms to ensure uniqueness.
- The desire in RL to keep updating representations as policies improve, preventing mismatch between the learned embedding and the actual transitions.

Hence, the authors build upon standard results for **Laplacian-based embedding** (Koren’s graph-drawing objective, Wu et al.’s approach for approximating the operator under a fixed policy, and Wang et al. and Gomez et al. for clarifying uniqueness/stability) and carry these ideas forward into **time-varying, policy-dependent** scenarios—bridging a gap in the prior literature where theoretical guarantees were lacking for fully online adaptation.

**Theoretical Claims:**

The central theorems establish that, under a bounded-drift assumption on the evolving policy, online projected gradient descent on the authors’ asymmetric graph-drawing objective converges (in an ergodic sense) to a critical point of the Laplacian representation. The paper’s proofs rely on two main ingredients:

1. **Equilibrium Characterization:**
   They show that if you hold the policy (and thus the underlying operator) fixed, then gradient descent on the proposed objective converges to a set of stable equilibrium points that match the smallest Laplacian eigenvectors. The argument resembles that of prior Laplacian objectives such as ALLO, but with “asymmetric” terms (including a stop-gradient operator) to break rotational symmetries.
   - The lemma mapping equilibrium points to eigenvectors (possibly permuted) follows an inductive argument on the dimension of each learned vector.
   - They then use a Jacobian-based stability analysis to show only the identity permutation is stable, and only if the vectors are all non-zero – which corresponds to the true eigenvectors in the correct order.

2. **Online Convergence:**
   They next show that with a policy that may change between steps but remains “close” from one update to the next (i.e. bounded drift), the difference in the objective from step to step is small enough to maintain progress. By combining a standard smoothness argument for gradient-based optimization with a drift analysis of the Markov chain, they give an ergodic bound on convergence to a stationary point.

On a closer look, the arguments generally follow a standard pattern in time-varying optimization:
- Proving each fixed-instance problem is well-behaved (smoothness, stable equilibria).
- Bounding the “error” introduced by drift in the environment’s operator.
- Applying a convergence result for online or dynamic projected gradient methods.

So far, the proofs appear internally consistent. Two subtle points might merit further scrutiny or clarification:

1. **Zero or Degenerate Eigenvectors**:
   Their analysis for “stable” equilibria relies on each eigenvector being non-zero (otherwise you could have fewer than \(d\) total eigenvectors). Although they show that only the full set is stable, it might be worth confirming if degenerate (partial) solutions could become temporarily stable under certain conditions in practice.

2. **Bounded-Drift Constant**:
   The drift assumption is crucial. The text references a sublinear bound on the cumulative changes to the policy. In practice, large updates might occur at times, especially before the policy converges. Whether the theorems still hold if the policy sometimes shifts drastically (but remains small in the long run) is an open question, though the paper’s arguments do rely on standard constraints from policy-gradient or trust-region methods.

Overall, the proofs for both static and dynamic cases align with prior lines of analysis in Laplacian representation learning and dynamic gradient-based optimization. There are no obvious errors in how they apply known stability and perturbation lemmas, and the reasoning steps seem coherent with the stated assumptions.

---

> ### Author Rebuttal · Authors · 2025-04-01
>
> We appreciate the reviewer's efforts in reviewing our work and providing their detailed feedback. Below we address their questions and concerns.
>
> **Zero Eigenvectors**
>
> We thank the reviewer for this insightful suggestion. Based on our analysis, we do not expect degenerate partial solutions to be stable. To further investigate this, we plan to conduct additional experiments by resetting one of the eigenvectors to near zero after a fixed number of training steps and observing how the representation evolves.
>
> **Bounded Drift**
>
> We thank the reviewer for their valuable suggestion. In Section 5 of the paper, we present empirical results for scenarios where the policy update may undergo rapid changes. Specifically, we present experimental results for Deep Q-Network (DQN), where the policy improvement step (i.e., greedy selection of actions) does not guarantee bounded policy drift. In Figure 4.1 (a) in which we compare the effect of bounded drift on the accuracy of the learned representation, Vanilla policy gradient (VPG) which might have drastic changes, and PPO (0.2, 0.01) follow a similar trend during the first $10^5$ steps. However, as PPO policy starts to stabilize the gap in accuracy increases at the end. This demonstrates that the total drift in addition to the drift limit matters more towards the accuracy of the representation quality in the long run as discussed by our assumption. In addition, we also compare the effect of rapid changes in the DQN algorithms. Through empirical studies, we conclude that the bounded drift assumption is necessary for learning an accurate representation. Additionally, we would like to highlight that the bounded policy drift is a desired feature of many RL algorithms in practice such as conservative policy iteration, PPO, and TRPO, which naturally limit such drastic shifts.
>
>
>
> **Contribution**
>
> To the best of our knowledge, this is the first work that addresses online Laplacian-based representation in reinforcement learning with theoretical guarantees. Our work provides the first analysis of the convergence of an online learning algorithm of the d-smallest eigenvectors of the Laplacian while updating the policy. Moreover, we provide an analysis for the convergence of AGDO (which is equivalent to ALLO with beta = 0) to the d-smallest eigenvectors of the Laplacian which uses slightly different steps than the work by Gomez et al(2023). Indeed, the work by Gomez et al.(2023) has been influential for our work, as it showed that optimizing the ALLO objective has the d-smallest eigenvectors as the only stable equilibrium for a fixed policy. We clarify the following in Section 4.2 of the paper: "This result is similar to Lemma 2 derived by Gomez et al.(2023) with the norm of the vectors being different and the fact that the vectors can be zero". However, we show in Theorem 4.4 that only the identity permutation with non-zero vectors corresponds to a stable equilibrium under a proper selection of hyperparameters. Additionally, our convergence analysis of the Laplacian representation in the online setting presents a novel perspective that, to our knowledge, has not been explored in prior work.
>
>
>
> **Extention to Large-scale Function Approximator and Continuous Control**
>
> We fully agree with the review that further experiments showcasing the value of our framework in RL algorithms would strengthen the importance of our contributions. The contributions of the current work are more of a theoretical nature, laying the theoretical foundation for provable online Laplacian representation learning which is accompanied by simulations that support the accuracy of the learned representation. Building upon the developed theory, we will consider applications of the learned representation in options learning in future work.
>
> Since our approach is built on standard online SGD and representation learning frameworks, it can be easily integrated in larger function approximators, going beyond MLPs over low-dimensional inputs. For example, the empirical study by Klissarov and Machado (2023) utilizes a convolutional neural network to learn the Laplacian representation in an online manner and uses the representation to learn options with high coverage.

---

### Official Review · Reviewer_2pDa · 2025-03-12

**Overall Recommendation:** 2

**Summary:**

The paper titled "Online Laplacian-Based Representation Learning in Reinforcement Learning" explores the integration of Laplacian-based representation learning within reinforcement learning (RL) frameworks. The authors address the challenge of learning Laplacian-based representations online and with theoretical guarantees along with policy learning. They propose an online optimization approach using the Asymmetric Graph Drawing Objective (AGDO) and demonstrate its convergence through online projected gradient descent.
The authors introduce AGDO, a simplified version of the Augmented Lagrangian Laplacian Objective (ALLO), and provide theoretical bounds for its convergence. The empirical analysis evaluates the accuracy of the proposed method in both fixed and online settings. Moreover, the experiments demonstrate that AGDO achieves performance comparable to ALLO. Finally, the authors analyze the impact of different components, such as bounded drift, number of update steps, and replay buffer size, on the accuracy of the learned representations.

The main contributions of the paper can be summarised as follows:
 - The authors introduce the Asymmetric Graph Drawing Objective (AGDO), a simplified version of ALLO that eliminates the need for dual variables.
 - They provide theoretical bounds for the convergence of AGDO under gradient descent dynamics.
 - The paper includes extensive simulation studies that empirically validate the convergence guarantees to the true Laplacian representation.
- The authors provide insights into the compatibility of different reinforcement learning algorithms with online representation learning.

**Claims And Evidence:**

The claims are generally supported by the evidence (with a few exceptions)

Well-supported claims:

1. Introduction of AGDO:
 - The paper introduces the Asymmetric Graph Drawing Objective (AGDO) as a simplified version of the augmented Lagrangian Laplacian objective (ALLO). This claim is supported by a detailed formulation of AGDO and its comparison with ALLO.
2. Theoretical analysis:
 - The authors provide theoretical bounds for the convergence of AGDO under gradient descent dynamics. This claim is supported by rigorous mathematical proofs and assumptions laid out in the paper, particularly in sections 4.2 and 4.3.
3. Empirical validation:
 - The paper includes extensive simulation studies that empirically validate the convergence guarantees to the true Laplacian representation. Figures 2 and 3 show the average cosine similarity between the true Laplacian representation and the learned representation using AGDO and ALLO, demonstrating comparable performance.
4. Compatibility insights:
 - The authors provide insights into the compatibility of different reinforcement learning algorithms with online representation learning. This claim is supported by empirical analysis and ablation studies, as shown in Figure 4, which analyze the impact of bounded drift, number of update steps, and replay buffer size on the accuracy of the learned representation.


Questions on other claims:

1. Bounded drift assumption:
 - The paper assumes that the drift in the policy caused by the policy learning algorithm is bounded. While this assumption is valid for many policy learning algorithms, the empirical evidence provided (Figure 4a) shows that the accuracy of the learned representation can vary significantly depending on the drift bound. The claim that bounded drift is necessary for learning an accurate representation is supported, but the variability in accuracy may suggest that further investigation is needed to fully understand the impact of different drift bounds.

2. Effect of number of update steps:
 - The paper claims that increasing the number of update steps per sample collected should enhance accuracy. However, the empirical findings indicate that this is not always observed, possibly due to noise caused by sampling from the replay buffer. This discrepancy suggests that the claim may need further empirical validation and refinement.

**Essential References Not Discussed:**

No

**Experimental Designs Or Analyses:**

Please, find below comments concerning the soundness and validity of the experimental designs and analyses presented in the paper as well as some issues

1. Grid world environments:
 - The use of grid world environments for testing the proposed methods is appropriate. These environments provide a controlled setting to evaluate the accuracy and convergence of the learned representations. The design includes varying the complexity of the environments, which helps in assessing the robustness of the methods.
2. Comparison with ALLO:
 - The experiments compare the performance of AGDO with ALLO in both fixed and online settings. This comparison is valid and provides a clear benchmark to evaluate the effectiveness of AGDO. The use of average cosine similarity as a metric is appropriate for measuring the alignment between the true and learned representations.
3. Ablation studies
 - The ablation studies analyze the impact of bounded drift, number of update steps, and replay buffer size on the accuracy of the learned representation. These factors are directly relevant to the performance of the proposed methods, making the experimental design comprehensive and insightful.
4. Theoretical analysis:
 - The paper provides a detailed theoretical analysis of the convergence of AGDO under gradient descent dynamics. The assumptions and mathematical proofs are clearly laid out, supporting the validity of the theoretical claims.
5. Empirical validation:
 - The empirical results are presented with average cosine similarity metrics and confidence intervals, which provide a clear and statistically sound measure of performance. The results show that AGDO achieves performance comparable to ALLO, supporting the claim of its effectiveness.

6. Bounded drift assumption - see Claims And Evidence
7. Effect of number of update steps - see Claims And Evidence
8. Replay buffer size:
 - The impact of replay buffer size on the accuracy of the learned representation shows that both very small and very large buffer sizes can negatively affect performance. This highlights the importance of carefully tuning the buffer size to balance variance and bias in the loss estimate.

**Methods And Evaluation Criteria:**

Yes, the proposed methods and evaluation criteria in the paper make sense for the problem and application at hand.

**Other Comments Or Suggestions:**

See Other Strengths And Weaknesses

**Other Strengths And Weaknesses:**

**Strengths**

1. Innovative approach:
 - The introduction of the Asymmetric Graph Drawing Objective (AGDO) as a simplified version of the augmented Lagrangian Laplacian objective (ALLO) is innovative. It addresses the complexity of dual variables, making the method more practical for online learning scenarios.
2. Comprehensive empirical validation:
 - The extensive simulation studies in both fixed and online settings provide strong empirical evidence for the effectiveness of AGDO. The use of average cosine similarity as a metric is appropriate and provides clear insights into the performance of the learned representations.
3. Compatibility insights:
 - The paper offers valuable insights into the compatibility of different reinforcement learning algorithms with online representation learning. This is particularly useful for practitioners looking to integrate AGDO with various RL frameworks.
4. Ablation studies:
 - The ablation studies analyzing the impact of bounded drift, number of update steps, and replay buffer size are insightful. They help in understanding the factors that influence the accuracy and performance of the learned representations.


**Weaknesses**

1. Bounded drift assumption:
 - The paper assumes that the drift in the policy caused by the policy learning algorithm is bounded. While this assumption is valid for many policy learning algorithms, the empirical evidence provided (Figure 4a) shows that the accuracy of the learned representation can vary significantly depending on the drift bound. The claim that bounded drift is necessary for learning an accurate representation is supported, but the variability in accuracy may suggest that further investigation is needed to fully understand the impact of different drift bounds.

2. Effect of number of update steps:
 - The paper claims that increasing the number of update steps per sample collected should enhance accuracy. However, the empirical findings indicate that this is not always observed, possibly due to noise caused by sampling from the replay buffer. This discrepancy suggests that the claim may need further empirical validation and refinement.


3. Replay buffer size:
 - The impact of replay buffer size on the accuracy of the learned representation shows that both very small and very large buffer sizes can negatively affect performance. This highlights the importance of carefully tuning the buffer size to balance variance and bias in the loss estimate.

5. Limited real-world validation:
 - The experiments are conducted in grid world environments, which, while useful for controlled testing, may not fully capture the complexities of real-world RL scenarios. Further validation in more complex and realistic environments would strengthen the findings.

6. Complexity of theoretical analysis:
 - The theoretical analysis, while rigorous, is quite complex and may be challenging for readers without a strong mathematical background. Simplifying some of the explanations or providing more intuitive insights could make the paper more accessible.

**Questions For Authors:**

See Other Strengths And Weaknesses

**Relation To Broader Scientific Literature:**

The key contributions of the paper are closely related to several prior findings, results, and ideas in the broader scientific literature.

Related findings/results/ideas:

1. Proto-Value functions:
 - The concept of proto-value functions introduced by Mahadevan (2005) is foundational to Laplacian-based representation learning. Proto-value functions are eigenfunctions of the normalized Laplacian of the graph generated by a random walk over the state space. This idea has influenced subsequent work on Laplacian representations, including the current paper's focus on eigenvectors of the Laplacian matrix.
2. Graph drawing objective:
 - The graph drawing objective proposed by Koren (2005) has been a key method for approximating Laplacian representations in large state spaces. Wu et al. (2018) extended this method to continuous state spaces, demonstrating its applicability through stochastic optimization. The current paper builds on these ideas by introducing AGDO and validating its performance.
3. Successor features:
 - The deep successor representation introduced by Kulkarni et al. (2016) decomposes the value function into a successor feature function and a reward predictor function. This approach has found applications in transfer learning and options discovery. The current paper's focus on Laplacian-based representation learning is related to these ideas, as both methods aim to improve RL algorithms' performance through better state representations.
4. Contrastive learning:
 - Contrastive learning methods, such as those proposed by Laskin et al. (2020) and Stooke et al. (2021), aim to learn representations that distinguish between similar and dissimilar pairs of data points. These methods are closely related to Laplacian-based representation learning, as both approaches seek to capture the geometry of the underlying state space. The current paper's empirical validation of AGDO aligns with these prior findings.

**Theoretical Claims:**

No

---

> ### Author Rebuttal · Authors · 2025-04-01
>
> We thank the reviewer for their feedback and addressed their questions and concerns below.
> > Bounded drift assumption
>
> Regarding the concern about drastic or erratic policy shifts in real-world RL scenarios, our theoretical analysis explicitly assumes bounded policy drift (Assumption 4.2). This is a desired feature of many RL algorithms in practice such as conservative policy iteration, PPO, and TRPO, which naturally limit such drastic shifts. Our convergence results hold under this assumption. Whether this assumption could be relaxed or modified for algorithms not promoting bounded drift would be an important and interesting future investigation. We acknowledge that handling larger or unpredictable policy changes would enhance practical robustness. Incorporating adaptive step-size methods or robust regularization techniques to explicitly avoid large policy deviations is a promising direction for future extensions of our framework.
>
> > Effect of number of update steps
>
> We would like to clarify that we do not make theoretical claims in the paper for the effect of the number of update steps on the performance. However, we empirically study the effect on the number of update steps and hypothesize that the discrepancy in the expected behavior is due to the presence of noise, caused by sampling from the replay buffer. We discuss this in detail in Section 5 (lines 430-436) as follows: "In Figure 4b, we analyze the effect of increasing the number of steps. We vary the number of update steps per sample between 1 and 20. While an increase in the number of steps is expected to enhance accuracy, our findings indicate that this is not observed. We hypothesize that this discrepancy is  due to the presence of noise, caused by sampling from the replay buffer."
>
> > Replay buffer size
>
> We absolutely agree with the reviewer's insight. In fact, we highlight this in Section 5 (lines 401-411) as follows: "In the online setting, the buffer would include steps from previous policies with different stationary and transition distributions which would introduce bias to our loss estimate. However, a small buffer size would also increase the variance of the estimate. This is confirmed by the results, as for a buffer that holds only one episode we see a worse performance than a buffer that holds 20 episodes. On the other hand, increasing the buffer size drastically also causes accuracy to drop as the samples used have a different distribution which can be seen for buffers with sizes 50 and 400". Hence, a careful balance of the buffer size is important to balance the variance and bias in the loss estimates.
>
> > Limited real-world validation
>
> We fully agree with the reviewer that further experiments showcasing the value of our framework in RL algorithms would strengthen the importance of our contributions. The contributions of the current work are more of a theoretical nature, laying the theoretical foundation for provable online Laplacian representation learning which is accompanied by simulations that support the accuracy of the learned representation. Building upon the developed theory, we will consider applications of the learned representation in options learning in future work.
>
> > Complexity of theoretical analysis
>
> This work builds upon an existing line of work that uses rigorous mathematical methods that provide both a practical application through empirical intuitive evaluations in addition to theoretical guarantees. The major contributions of this work are more of a theoretical nature, and given the complexities of joint representation learning and reinforcement learning, unfortunately, the theoretical analysis and results are unavoidably involved. Overall, our work offers rigorous proofs for the involved reader and in addition empirical results along with practical discussions and evaluations to provide some intuition of our theoretical findings.

---

### Official Review · Reviewer_XDtX · 2025-03-14

**Overall Recommendation:** 4

**Summary:**

The paper is about representation learning in RL. It analyses an online algorithm designed to learn Laplacian representations of a Markov Decision Process (MDP). Given a fixed policy $\pi$, generally the exploratory policy, the Laplacian operator is defined as $L = I - \frac{P_{\pi} + P_{\pi}^{\ast}}{2}$, where $P_{\pi}$ is the transition matrix of the Markov chain on states induced by $\pi$, and $P_{\pi}^{\ast}$ is an adjoint to $P_{\pi}$ with respect to some underlying measure. The Laplacian representation then consists in computing the first eigenvectors of L, for instance to perfom linear value function approximation. To avoid direct computation of the eigenvectors of $L$, earlier works have proposed optimzation procedures based on minimizing Rayleigh quotients: $\min_{u_i} \sum_{i=1}^{d} u_i^{\top} L u_i$ as the minimum would be the sum of the first $d$ eigenvalues $\sum_{i=1}^{d} \lambda_i$, achieved for the d first eigenvectors of $L$. The specificity of this paper is to consider a setting where the policy $\pi$ is no longer fixed but evolves in time, and learning of the Laplacian representation needs to be performed in an online manner while the policy evolves. This is motivated by earlier works which show empirically that learning Laplacian representations during exploration could improve performance. This paper aims to bring a theoretical counterpart to this work. The authors propose  thus a new optimzation objective, AGDO,  to perform online computation of the Laplacian representation, and study its convergence properties. In a first theorem, they show that under gradient descent dynamics, with a good choice of the parameters, the only stable equilibrium points correspond to the true Laplacian eigenvectors. Secondly, they propose an online projected gradient descent (OPGD), which under a drift assumption on the policy process, is proved to converge towards this equilibrium. This theoretical work is completed with numerical experiments.

**Claims And Evidence:**

I haven't given a thorough look at the proofs in the appendix, but this work appears as serious to me, the authors provide proofs for all their claims. It builds on prevous works about computing Laplacian eigenvectors through minimization objective and adapts them to the online setting.

**Essential References Not Discussed:**

In my opinion the paper already cites the most relevant references.

**Experimental Designs Or Analyses:**

The content of the paper is mostly theoretical, and I have nothing to complain about when it comes to the experimental work.

**Methods And Evaluation Criteria:**

The AGDO objective and optimization methods used in the paper build on a series of work on Laplacian representations intitiated with the paper of Koren 2005. In that regard I think the algorithm is well motivated. It could be considered a limitation however that the experiments limit to gridworld and maze environments.

**Other Comments Or Suggestions:**

I don't have a lot of suggestions for the authors. They mention however successor representations in the related work section, so I wondered if maybe their work and method could also be related with the work "Does Zero-Shot Reinforcement Learning Exist?" by Touati \& al ? In the latter and the references therein, the authors propose a "forward-backward" representation of a RL problem which is based on a low-rank approximation of the successor measure. I think there could be a common point with the present paper in that learning their model requires updating the policies and the representation at the same time.

**Other Strengths And Weaknesses:**

The paper provides both a theoretical and empirical study of the online learning of Laplacian representations. It is well presented and clear.

**Questions For Authors:**

I dont't have a big concern or question whose answer would heavily affect my opinion on the paper.

**Relation To Broader Scientific Literature:**

Laplacian representations have been a recurring theme in RL during the last 20 years. This paper is mostly concerned with the numerical aspects and algorithms that compute eigenvectors while avoiding matrix computations. The basic idea appeared in a work by Koren 2005 who proposed computing eigenvectors exploiting the fact they minimize Rayleigh quotients. Since then a series of work have improved on this approach, notably to address symmetry issues resulting in non-uniqueness of the solutions for the minimization objectives. The present paper draws inspiration from the most recent work in that aspect (Gomez \& al. 2023), to adapt the idea to an online version of the problem. This is motivated by a paper of Klissarov \& Machado 2023, which showed experimentally that improvements could be made by updating jointly the policy and the representation.

**Theoretical Claims:**

Due to a lack of time I haven't checked completely the proofs of the theoretical claims.

---

> ### Author Rebuttal · Authors · 2025-04-01
>
> We sincerely appreciate the reviewer's thoughtful and considerate comments. Below we address their remark about related work.
>
> > I don't have a lot of suggestions for the authors. They mention however successor representations in the related work section, so I wondered if maybe their work and method could also be related with the work "Does Zero-Shot Reinforcement Learning Exist?" by Touati \& al ? In the latter and the references therein, the authors propose a "forward-backward" representation of a RL problem which is based on a low-rank approximation of the successor measure. I think there could be a common point with the present paper in that learning their model requires updating the policies and the representation at the same time.
>
> We appreciate the reviewer's suggestion; this paper is indeed relevant to our work. Touati et al. explore approximate zero-shot reinforcement learning, where an agent first learns environment features in a reward-free phase and then deploys a learned linear approximation of the reward function. Their study compares two approaches:
>
> - Successor Features Representation – Elementary features are learned offline, followed by successor feature learning.
>
> - Forward-Backward Representation – Both elementary and successor features are learned simultaneously.
>
> Their results highlight that Laplacian-based representations outperform other feature choices. Furthermore, the dynamic approach of jointly learning elementary and successor features proves to be superior.
>
> This work underscores the importance of high-quality representation learning, the effectiveness of Laplacian-based representations, and the benefits of jointly learning representations. We will incorporate this discussion into the successor feature paragraph in Section 2.

---

### Official Review · Reviewer_H35G · 2025-03-14

**Overall Recommendation:** 3

**Summary:**

The authors propose a theoretical analysis of an algorithm for online representation learning based on Laplacian eigenvectors. Current methods tend to learn representations based on a uniform/exploratory dataset, while the authors argue that representations evolve with policy changes and that more efficient learning can be achieved by considering this. Their algorithm is a simple modification of an established algorithm called ALLO. They prove that it is possible to learn representations efficiently, provided that policies do not change too rapidly (under bounded policy drift).

**Claims And Evidence:**

Yes.

**Essential References Not Discussed:**

It would be helpful if the authors could confirm whether this is indeed the first paper to provide guarantees for online representation learning, as this does not seem very likely to me.

**Experimental Designs Or Analyses:**

No, I have not.

**Methods And Evaluation Criteria:**

I wish the experimental section were more extensive. The whole premise of the paper is that learning representations online provides an advantage over learning a fixed representation using a uniform policy. However, I do not see clear evidence of this in the experimental section.

**Other Comments Or Suggestions:**

- I am not sure what current overall policy in line 078 left means.
- DCEO acronym in line 118 left is not explained
- line 167 left, should be "second to the smallest eigenvalue"?
- what is $\Gamma$ in line 257 left?
- Eq (9) should the last term be "2b..." or just "b..."?
- Eq (10) should be $G_{u_i}^{(t)}(u^{(t)})$, then in line 280 again the same thing, and inside of the definition first $u_i$ should be $u_i^{(t)}$ I believe.
- I am not sure that $L^{(t+1)}_{\rho^{(t+1)}}$ in (c) in Lemma 4.6 has been defined previously. Also missing brackets for $t+1$ and $t$ superscripts in (d).
- In Algorithm 1, line 5: should it be using $\mathcal{A}$ and $u^{t+1}$?
- Has $G^{(t)}$ in Theorem 4.7 been defined somewhere, since you used $G^{(t)}_{u_i}$ before?
- line 372 right, last ALLO should be AGDO probably.
- line 407 left, ":" instead of ";"

**Other Strengths And Weaknesses:**

I believe the question the authors aim to address is both important and challenging. The proposed method appears reasonable and applicable to realistic scenarios. The theoretical analysis is interesting, as it highlights the importance of bounded policy drift for efficient representation learning.

On the weaker side, I find the experimental section rather unconvincing. Additionally, I am unsure about the novelty of the methodological contribution in this paper.

**Questions For Authors:**

1. The fact that you can effectively learn representations of current policies does not necessarily imply much about learning a good policy, right? How does the error in your representation learning algorithm influence the policy update step? In other words, do you have any insight into the stationary point your algorithm converges to?

2. Could you please comment on the purpose of comparing AGDO and ALLO? Based on the experimental results, they appear nearly identical, and it is not clear to me why solving ALLO would be more difficult.

3. In Section 2, you mention that previous work has shown the online version of representation learning performs comparably to a two-stage variant of the algorithm. Have you conducted similar experiments for your algorithm? If so, and if the results are similar, what are the benefits of online learning?

**Relation To Broader Scientific Literature:**

I believe the authors did a good job of providing an extensive literature review in Section 2, presenting a long line of work on learning Laplacian representations alongside a more recent perspective on representation learning using neural networks.

**Theoretical Claims:**

No, I did not check the correctness of the proofs.

---

> ### Author Rebuttal · Authors · 2025-04-01
>
> We would like to thank the reviewer for taking the time to give a thorough review. We address their questions and concerns below. Due to space constraints, we have shortened the quotes from the review. We also appreciate the reviewer’s suggestions in the "Other Comments or Suggestions" section and will address typos and clarifications in the final version or during the discussion phase if the reviewer provides a rebuttal comment.
>
> **Addressing the impact of learning the Laplacian representation online**
> > The fact that you ... imply much about learning a good policy, right?
>
> Indeed, this is a valuable insight. The accuracy of the learned representation does not necessarily imply a good policy.
>
> > How does the error .. converges to?
>
> An accurate MDP representation improves sample efficiency and average returns, as shown in options learning (Chen et al., 2024), reward shaping (Wu et al., 2018), and linear function approximators for actor-critic methods (Mahadevan & Maggioni, 2007). While Laplacian representation learning typically relies on a fixed uniform policy, we argue that online learning can be more effective in scenarios with concentrated initial distributions. Additionally, our online learning method can be easily extended to non-stationary environments with a bounded drift. Finally, our method avoids extra pre-training samples by leveraging those already collected for policy training.
>
> Regarding the stationary point of the policy, depending on the type of reinforcement learning algorithm and the way it uses the representation, the stationary point of the algorithm could differ. For example, in an option learning method, it could correspond to multiple options with high coverage. In a reward-shaping goal-based environment, the stationary point could be a policy that reaches the goal. In our work, we assume that the policy and representation are learned in parallel and that the policy learning algorithm converges to some fixed point. The policy does not necessarily use the learned Laplacian features as inputs, instead the Laplacian representation can influence learning in other ways such as reward shaping or exploration rewards used in options learning.
>
> > In Section 2, you mention that previous ... learning?
>
> We appreciate the reviewer raising this important point. We have not specifically tested the application of online representation learning for options learning. However, Klissarov and Machado’s work demonstrates that when the Laplacian representation is used solely for learning an exploration reward in an options learning framework and is used exclusively for exploration, both the online and fixed representations yield similar performance. While our current focus has not included this specific comparison, we plan in future work to explore broader applications of the learned representation, including value function approximation, reward shaping, and option learning frameworks that address both exploration and diversity. Additionally, we aim to investigate frameworks where learned options are actively utilized during the execution phase.
>
> **Contribution and Experimental Section**
>
> We would like to confirm that to the best of our knowledge, this is the first work that addresses online Laplacian-based representation in reinforcement learning with theoretical guarantees. Our work provides the first analysis of the convergence of an online learning algorithm to the d-smallest eigenvectors of the Laplacian while updating the policy. Moreover, for a fixed policy, we provide an analysis for the convergence of AGDO to the d-smallest eigenvectors of the Laplacian which uses slightly different steps than the work by Gomez et al. However, we do not claim to be the first in online representation learning overall, as other methods, like successor features (discussed in Section 2), also learn policy and representation simultaneously.
>
> We fully agree with the reviewer that further experiments showcasing the value of our framework in RL algorithms would strengthen the importance of our contributions. The contributions of the current work are more of a theoretical nature, laying the theoretical foundation for provable online Laplacian representation learning which is accompanied by simulations that support the accuracy of the learned representation. Building upon the developed theory, we will consider applications of the learned representation in options learning in future work.
>
> **Comparing AGDO and ALLO**
>
> Since AGDO is a simplified version of ALLO as it does not require an extra update step for the dual parameters, we empirically verify our theoretical analysis that AGDO, similar to ALLO, has the d-smallest eigenvectors as the only stable equilibrium. The results verify our claims. The reason they appear identical could stem from the fact that AGDO can be viewed as a regularized version of ALLO with a closed-form solution to the dual parameters as we discuss in section 4.1.

---

### Decision · Program_Chairs · 2025-05-01

**Decision:**

Accept (poster)

**Comment:**

The reviews generally lean towards acceptance for providing valuable contributions to an well-studied area of representation learning for RL, with some weakness around the limitations of the experimental work. I tend to agree with both of these points, but also want to highlight a comment made by the author(s) that could be revisited.

In response to a reviewer question: “The fact that you can effectively learn representations of current policies does not necessarily imply much about learning a good policy, right?”

The author(s) indicate “The accuracy of the learned representation does not necessarily imply a good policy.”

There are results in approximate policy iteration (e.g. Error bounds for approximate policy iteration. Munos. 2003) and elsewhere that could be used to argue that (stated informally) having representations that accurately represent the value function for policies throughout policy improvement leads to improved bounds on the quality of the policy you converge to. So, while having effective representations for current policies does not *imply* a better policy, it could improve the upper bound on the eventual learned policy.